# BiAssemble: Learning Collaborative Affordance for Bimanual Geometric Assembly

**Yan Shen** [* 1 2]  **Ruihai Wu** [* 1]  **Yubin Ke** [1]  **Xinyuan Song** [1]  **Zeyi Li** [1]
**Xiaoqi Li** [1 2]  **Hongwei Fan** [1 2]  **Haoran Lu** [1]  **Hao Dong** [1 2]

## Abstract

Shape assembly, the process of combining parts into a complete whole, is a crucial robotic skill with broad real-world applications. Among various assembly tasks, geometric assembly—where broken parts are reassembled into their original form (*e.g.*, reconstructing a shattered bowl)—is particularly challenging. This requires the robot to recognize geometric cues for grasping, assembly, and subsequent bimanual collaborative manipulation on varied fragments. In this paper, we exploit the geometric generalization of point-level affordance, learning affordance aware of bimanual collaboration in geometric assembly with long-horizon action sequences. To address the evaluation ambiguity caused by geometry diversity of broken parts, we introduce a real-world benchmark featuring geometric variety and global reproducibility. Extensive experiments demonstrate the superiority of our approach over both previous affordance-based and imitation-based methods. Project page: https://sites.google.com/view/biassembly/.

## 1. Introduction

Shape assembly, the task of assembling individual parts into a complete whole, is a critical skill for robots with wide-ranging real-world applications. This task can be broadly categorized into two main branches: furniture assembly (Zhan et al., 2020; Heo et al., 2023; Lee et al., 2021) and geometric assembly (Wu et al., 2023c; Sellán et al., 2022; Lu et al., 2024c). Furniture assembly focuses on combining functional components, such as chair legs and arms, into a fully constructed piece, emphasizing both the functional role of each part and the overall structural design. In contrast, geometric assembly involves reconstructing broken objects, like piecing together parts of a shattered mug, to restore their original form. While furniture assembly has been relatively well-studied—ranging from computer vision tasks that predict part poses in the assembled object (Zhan et al., 2020) to robotic systems that assemble parts in both simulation (Ankile et al., 2024; Yu et al., 2021; Wang et al., 2022a) and real-world environments (Heo et al., 2023; Suárez-Ruiz et al., 2018; Xian et al., 2017)—geometric assembly remains under-explored despite its significant potential for real-world applications (Sellán et al., 2022; Lu et al., 2024b), such as repairing broken household items, reconstructing archaeological artifacts (Papaioannou & Karabassi, 2003), assembling irregularly shaped objects in industrial tasks, aligning bone fragments in surgery (Liu et al., 2014), and reconstructing fossils in paleontology (Clarke et al., 2005).

Previous works on geometric assembly primarily focused on generating physically plausible broken parts through precise physics simulations in the graphics domain (Sellán et al., 2022; 2023), and estimating the target assembled part poses based on observations in the computer vision domain (Wu et al., 2023c; Lu et al., 2024c; Qi et al., 2025). These studies only consider the geometries and ideal assembled poses of broken parts, dismissing the process of step-by-step assembling parts to the complete shape. However, different from opening a door or closing a drawer, only with the ideal part poses, it is difficult for a robot to directly and successfully manipulate broken parts to the complete shape.

The challenges of the above robotic geometric shape assembly task mainly come from the exceptionally large observation and action spaces. For the observation space, the broken parts have arbitrary geometries, and the graspness on the object surface should consider not only the local geometry itself, but also whether grasping on such point can afford the subsequent bimanual assembly actions. For the action space, as illustrated in Figure 1, it requires long-horizon action trajectories. Given the contact-rich nature of the task, where collisions among the two parts and two robots will easily exist, the actions should be fine-grained and aware

---

[*]Equal contribution [1]CFCS, School of Computer Science, Peking University [2]PKU-Agibot Lab. Correspondence to: Hao Dong <hao.dong@pku.edu.cn>.

*Proceedings of the 42nd International Conference on Machine Learning*, Vancouver, Canada. PMLR 267, 2025. Copyright 2025 by the author(s).

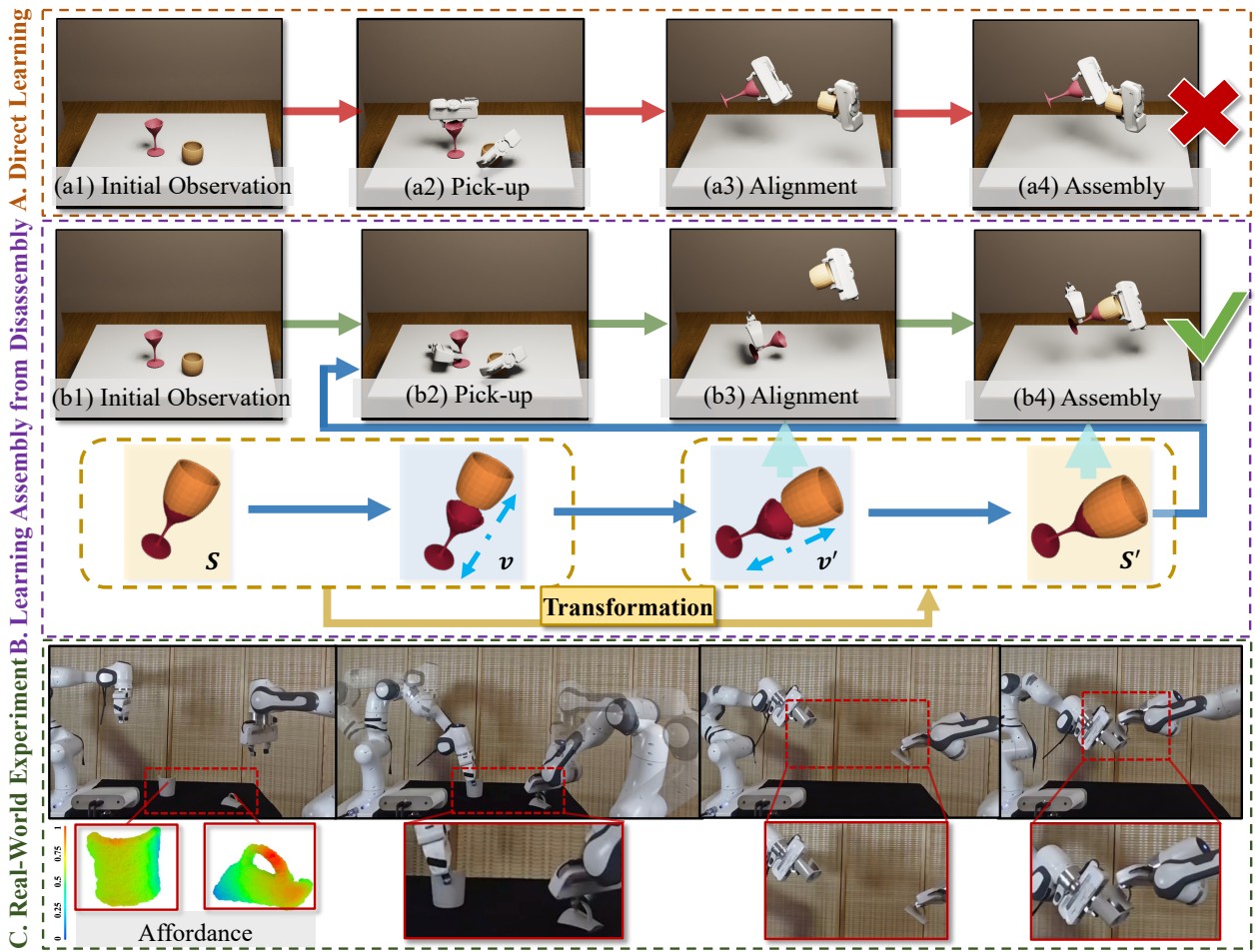

Figure 1. (**A**) Direct learning long-horizon action trajectories of geometric assembly may face many challenges: grasping ungraspable points, grasping points not suitable for assembly (*e.g.*, seams of fragments), robot colliding with parts and the other robot. (**B**) We formulate this task into 3 steps: pick-up, alignment and assembly. For assembly, we predict the direction that will not result in part collisions. For alignment, we transformed any assembled poses to poses easy for the robot to manipulate from the initial poses without collisions. For pick-up, we learn point-level affordance aware of graspness and the following 2 steps. (**C**) Real-World Evaluations with affordance predictions on two mugs and the corresponding manipulation.

of bimanual collaboration. Consequently, the policy must account for geometry, contact-rich assembly processes, and bimanual coordination.

We propose our **BiAssemble** framework for this challenging task. For geometric awareness, we utilize point-level affordance, which is trained to focus on local geometry. This approach has demonstrated strong geometric generalization in diverse tasks (Wu et al., 2022; 2023b), including short-term bimanual manipulation (Zhao et al., 2022), such as pushing a box or lifting a basket. To enhance the affordance model with an understanding of subsequent long-horizon bimanual assembly actions, we draw inspiration from how humans intuitively assemble fragments: after picking up two fragments, we align them at the seam, deliberately leaving a gap (since directly placing them in the target pose often causes

geometric collisions), with part poses denoted as alignment poses. We then gradually move the fragments toward each other to fit them together precisely. The alignment poses of the two fragments can be obtained by disassembling assembled parts in opposite directions. With this information, it becomes straightforward to extend the geometry-aware affordance to further be aware of whether the controller can move fragments into the alignment poses without collisions.

We develop a simulation environment where robots can be controlled to assemble broken parts. This simulation environment bridges the gap between vision-based pose prediction for broken parts and the real-world robotic geometric assembly. Moreover, since broken parts exhibit varied geometries (*e.g.*, the same bowl falling from different heights breaking into different groups of fragments), it is challeng-

ing to fairly assess policy performance in real-world settings. To address this, we further introduce a real-world benchmark featuring globally available objects with reproducible broken parts, along with their corresponding 3D meshes, which can be integrated into simulation environment. This benchmark enables consistent and fair evaluation of robotic geometric assembly policies. Extensive experiments on diverse categories demonstrate the superiority of our method both quantitatively and qualitatively.

## 2. Related Work

### 2.1. 3D Shape Assembly

Shape assembly is a well-established problem in visual manipulation, with many studies focusing on constructing a complete shape from given parts, typically involving the pose prediction of each part for accurate placement (Zhan et al., 2020; Wang et al., 2022a). Further work (Heo et al., 2023; Tian et al., 2022; Jones et al., 2021; Willis et al., 2022; Tie et al., 2025) studied assembly with robotic execution, requiring robots to carry out each step, with benchmarks spanning various applications from home furniture assembly (Lee et al., 2021) to factory-based nut-and-bolt interactions (Narang et al., 2022). We categorize these tasks into two types: furniture assembly and geometric assembly. This paper focuses on geometric assembly, where pieces are irregular and lack semantic definitions, such as in the case of a broken bowl, making categorization difficult. This contrasts with furniture assembly that involves parts like nuts, bolts, and screws, each with specific functions and clear categorization.

Previous work on geometric assembly (Sellán et al., 2022; Chen et al., 2022; Lu et al., 2024c), includes Wu et al. (2023c), which learns SE(3)-equivariant part representations by capturing part correlations for assembly, and Lee et al. (2024), which introduces a low-complexity, high-order feature transform layer that refines feature pair matching. However, these methods primarily focus on synthesizing parts into a cohesive object based on pose considerations without incorporating robotic execution, which is impractical in real-world scenarios where collisions may occur if the assembly process ignores actions. To tackle this challenge, we introduce a robotic bimanual geometric assembly framework that leverages two robots to collaboratively assemble pieces, enhancing stability in real-world execution.

### 2.2. Bimanual Manipulation

Bimanual manipulation (Chen et al., 2023; Grannen et al., 2023; Mu et al., 2021; Chitnis et al., 2020; Lee et al., 2015; Xie et al., 2020; Ren et al., 2024b; Liu et al., 2024; 2022; Li et al., 2023; Mu et al., 2024) offers several advantages, particularly in tasks requiring stable control or wide action

space. Current research primarily focuses on planning and collaboration. ACT (Fu et al., 2024; Zhao et al., 2023) introduces a transformer-based encoder-decoder architecture that leverages semantic knowledge from image inputs to predict bimanual actions. PerAct2 (Grotz et al., 2024) learns features at both voxel and language levels, utilizing shared and private transformer blocks to coordinate two robotic arms based on semantic instructions. However, in tasks rich in geometric complexity, where objects have limited semantic information but intricate geometric structures, these approaches—focused on semantic understanding—may encounter generalization limits. DualAfford (Zhao et al., 2022) learns point-level collaborative visual affordance, while only for short-term tasks like pushing or rotating. To address this, we leverage the geometric generalization capability of point-level affordance, and enhance it with the awareness of subsequent long-horizon assembly actions.

### 2.3. Visual Affordance for Robotic Manipulation

Among various vision-based approaches for robotic manipulation (Goyal et al., 2023; Brohan et al., 2023; Ze et al., 2024; Ju et al., 2024), for objects with rich geometric information and tasks requiring geometric generalization, point-level affordance, which reflects the functionality of each point for downstream manipulation (Mo et al., 2021; Li et al., 2024a), is broadly leveraged and can easily generalize to novel shapes with similar local geometries. A series of research have leverage this representation to a broad range of robotic manipulation tasks, such as deformable object manipulation (Wu et al., 2023b; Lu et al., 2024a; Wu et al., 2024), object manipulation in complex environments (Ding et al., 2024; Li et al., 2024b; Wu et al., 2023a), object manipulation with efficient exploration (Ning et al., 2024; Wang et al., 2022b), and short-term bimanual manipulation (Zhao et al., 2022). Leveraging the strengths of affordance representation, we design a sophisticated approach that incorporates this representation into bimanual geometric assembly task requiring long-horizon fine-grained actions, enhancing generalization and enabling more effective collaboration in addressing long-horizon geometric assembly challenges.

## 3. Problem Formulation

The task is to use two grippers to assemble a pair of 3D fractured parts initialized in random poses on the table. A camera situated in front of the table captures a partially scanned point cloud $O$. Given $O$, current state-of-the-art methods (Scarpellini et al., 2024; Lu et al., 2024c; Chen et al., 2022) can imagine the assembled part poses and the assembled object $S$ in any pose. Thus we assume taking imaginary assembled shape $S$ as the input.

For the policy $\pi$, as illustrated in Figure 1, we can simplify this long-term process into 3 key steps:

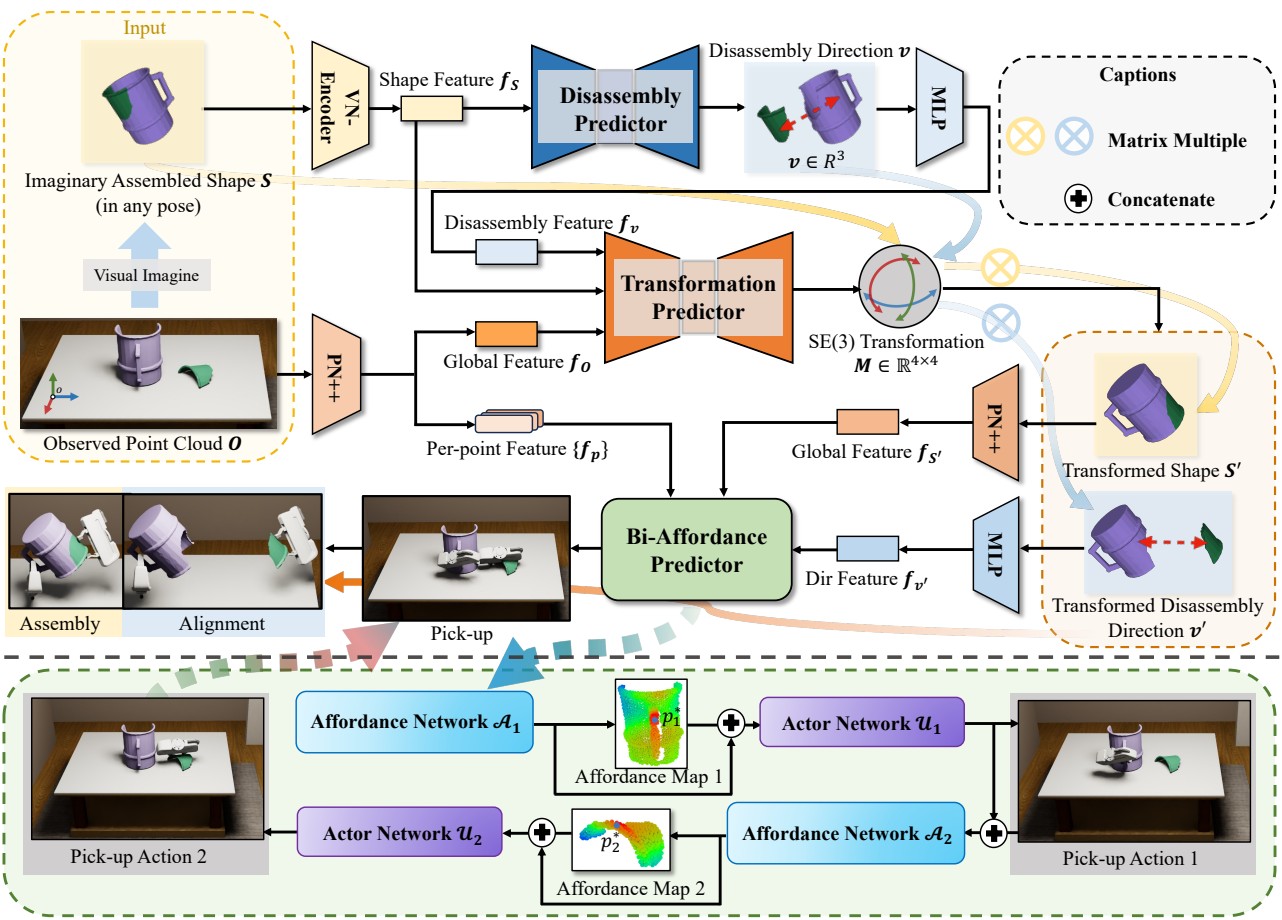

*Figure 2.* **Framework Overview.** With the point cloud observation and Imaginary Assembled Shape, the model predicts the disassembly direction in which the disassembled part poses can be easily reached by manipulating the raw parts under the guidance Bi-Affordance.

- **Pick-up**: the two grippers pick up the fractured parts with actions $(g_1^{pick}, g_2^{pick})$;
- **Alignment**: grippers carry parts to alignment poses with actions $(g_1^{align}, g_2^{align})$, positioning part seams to face each other and ensuring precise alignment for a perfect assembly;
- **Assembly**: grippers move forward to complete the assembly with actions $(g_1^{asm}, g_2^{asm})$.

Here, 1 and 2 denote the left and the right grippers, respectively. Each gripper action $g$ is formulated as an $SE(3)$ matrix, representing the gripper pose in 3D space.

## 4. Method

### 4.1. Overview

Our BiAssembly framework is designed to predict collaborative affordance and gripper actions for bimanual geometric shape assembly. First, to propose the assembly direction on two aligned parts, we develop the Disassembly Predic-

tor to learn the feasible disassembly directions in which the opposite assembly direction will result in no collisions, based on the fracture geometry of the imaginary assembled shape in any pose (4.2). Next, we design the Transformation Predictor, to transform disaasembled parts to poses where the controller can successfully manipulate the initial parts to these alignment poses (4.3). Based on the predicted part alignment poses, we propose the BiAffordance Predictor, which not only predicts where to grasp the fractured parts, but also considers the subsequent collaborative alignment and assembly steps (4.4). Finally, we explain training strategy and loss functions (4.6).

### 4.2. Disassembly Prediction Based on Shape Geometry

The set of feasible disassembly directions (in which the disassembly and opposite assembly processes will not result in collisions) is an inherent attribute of a pair of fractured parts, determined by fracture geometries. Therefore, we predict the disassembly directions, from the object-centric perspective, on the imaginary assembled shape $S$ in any

pose. Additionally, we observe that when fractured parts rotate, the feasible disassembly directions will rotate correspondingly, maintaining SO(3) equivariance relative to part poses. This SO(3) equivariance property is advantageous for disentangling shape geometry from shape poses, as demonstrated in previous works (Wu et al., 2023c; Scarpellini et al., 2024). Therefore, we adopt VN-DGCNN (Wu et al., 2023c; Deng et al., 2021) to encode the imaginary assembled shape parts $S$ and acquire the SO(3)-equivariant shape feature $f_s$.

Inputting the equivariant representation $f_s$, we use the Disassembly Predictor to predict the distribution of disassembly directions. Concretely, the Disassembly Predictor is implemented as a conditional variational autoencoder (cVAE) (Sohn et al., 2015), where the cVAE encoder maps the input disassembly direction $v$ into Gaussian noise $z \in \mathbb{R}^{32}$, and the cVAE decoder reconstructs the disassembly direction $v$ from $z$, with $f_s$ as the condition.

### 4.3. Transformation Prediction For Alignment Pose

Given the object-centric disassembly direction resulting in no collisions in the last-step assembly, we want to predict the alignment poses, where the robot can manipulate two parts from the initial poses to the alignment poses without collisions, and then robot can execute the assembly step. This problem can be formulated as predicting an SE(3) transformation $M \in \mathbb{R}^{4 \times 4}$ that is applied to the combination of the imaginary assembled shape $S$ and the disassembly direction $v$. To capture this, we adopt PointNet++ (Qi et al., 2017a;b) to encode the initial point cloud observation $O$ into the global feature $f_O$. We also employ an MLP to encode disassembly direction $v$ into feature $f_v$. The transformation predictor, which is implemented as a cVAE, takes in concatenation of $(f_O, f_v)$ to predict the SE(3) transformation $M$. The yellow and blue arrow in Fig. 2 illustrate the data flow in this process: by applying the transformation $M$ to the imaginary assembly $S$ and disassembly direction $v$, we obtain the transformed $S'$ and $v'$. Therefore, we can get the target poses to which the objects should be moved during the alignment and assembly phases.

### 4.4. BiAffordance Predictor

We build the BiAffordance Predictor to propose actions in the **Pick-up** step, indicating where to grasp the two fractured parts that can facilitate the whole long-horizon robotic assembly task. The BiAffordance Predictor should both identify easy-to-grasp regions on fractured parts and consider the subsequent **Alignment** and **Assembly** steps. This means (1) avoiding grasping regions in the seam and (2) preventing each gripper from adopting poses that could collide with the other part or gripper during subsequent steps.

Following DualAfford (Zhao et al., 2022), we disentangle the bimanual task into two conditional submodules. As presented in Fig. 2 (bottom), during inference, the BiAffordance Predictor conditionally predicts two gripper actions. The first Affordance Network generates the affordance map for the first gripper, highlighting the actionable regions for the bimanual assembly task, and we select a contact point $p_1^*$ with high actionable value. Then, the Actor Network predicts the gripper orientation $r_1$ for interaction at $p_1^*$. Based on the first action $g_1 = (p_1^*, r_1)$, we can then predict the second gripper action $g_2 = (p_2^*, r_2)$ using the second Affordance Network and Actor Network.

Different from DualAfford that only predicts affordance for short-term tasks, we use whether the manipulation points can satisfy the following alignment pose (by the robotic controller) and the assembly step as the training signal.

To encode input information, one PointNet++ encodes the initial point cloud $O$ obtains per-point features $\{f_p\}$. Another PointNet++ encodes the transformed shape $S'$ and derives the global feature $f_{s'}$. Additionally, an MLP encodes the transformed disassembly direction $v'$ into $f_{v'}$.

For Affordance and Actor Networks' designs, the first Affordance Network is implemented as an MLP that receives the concatenated features $(f_p, f_{S'}, f_{v'})$ and predicts an affordance score in the range of $[0, 1]$ for each point $p$. By aggregating the affordance scores of all points, we obtain the first affordance map, and from which we select $p_1^*$. The first Actor Network is implemented as a cVAE that takes the concatenated features $(f_{p_1^*}, f_{S'}, f_{v'})$ as condition, and outputs the gripper orientation $r_1$. The design of the second Affordance Network and Actor Network follows a similar structure, with the difference that they additionally incorporate the first gripper action's feature $(f_{p_1^*}, f_{r_1})$ along with $(f_p, f_{S'}, f_{v'})$. More details about the BiAffordance Predictor can be found in Appendix C.

### 4.5. Alignment and Assembly Actions

After successfully grasping a part, we now predict the gripper alignment poses $g_i^{align}$ and assembly poses $g_i^{asm}$, with $i \in \{1, 2\}$ denotes the gripper id. We assume the relative pose between the gripper and the object remains stable. For example, in the first pickup step and the third assembly step, the relative gripper-object pose remains consistent, as expressed in Equation 1:

$$g_i^{pick} \cdot q_i^{pick} = g_i^{asm} \cdot q_i^{asm}; \qquad g, q \in SE(3). \quad (1)$$

Here, $g$ and $q$ denote gripper and object poses, respectively.

Next, as we have the imaginary part shapes $\mathcal{P}$ with pose $q_i^{init}$, we can utilize a pretrained pose estimation model (Wen et al., 2024) to predict the relative pose of $q_i^{pick}$ with respect to $q_i^{init}$. Besides, by applying the predicted transformation $M$ to $\mathcal{P}$, we obtain the target assembled part $\mathcal{P}'$ and its pose as $q_i^{asm} = M \cdot q_i^{init}$. The gripper pose $g_i^{pick}$

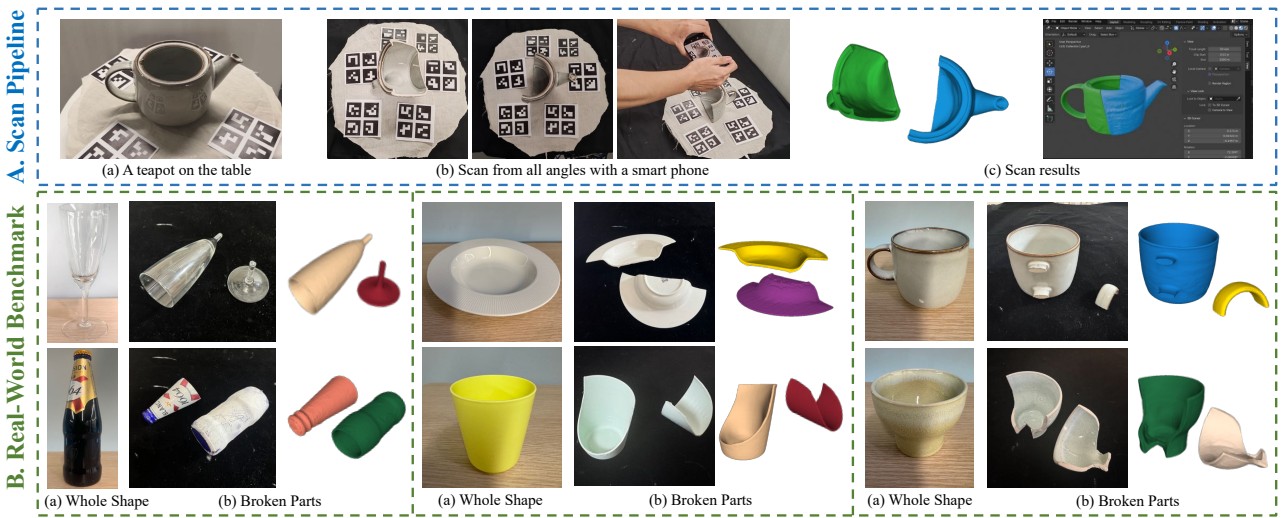

*Figure 3.* **Part A** illustrates the pipeline for scanning and reconstructing real objects. **Part B** presents examples of fractured parts from various categories, showcasing diverse geometries.

can be acquired from the robot control interface. Therefore, the gripper's final pose for assembling the parts can be calculated using Equation 2:

$$g_i^{asm} = g_i^{pick} \cdot q_i^{pick} \cdot (q_i^{init})^{-1} \cdot M^{-1}; \quad g, q \in SE(3). \quad (2)$$

It is important to note that, as indicated in the above simplified equation, we do not need to define a canonical pose or try to obtain the values of $q_i^{init}$; we only require the relative pose of $q_i^{pick}$ to $q_i^{init}$.

A similar relationship can be established between the first and the second intermediate steps, with the difference being that $q_i^{align} = M \cdot q_i^{init} + v'$.

### 4.6. Training and Losses

**Disassembly Direction Loss.** The Disassembly Predictor is implemented as cVAE. We apply Cosine Similarity Loss to measure the error between the reconstructed disassembly direction $v$ and ground-truth $v^*$, and KL Divergence to measure the difference between two distributions:

$$\mathcal{L}_D = \mathcal{L}_{CLS}(v, v^*) + D_{KL}\big(q(z|v^*, f_s)||\mathcal{N}(0,1)\big). \quad (3)$$

**Transformation Loss.** The predicted SE(3) transformation matrix $M$ consists of translation $T$ and rotation $R$. Our model predicts the translation as a 3D-vector using L1 Loss. The rotation, represented as a SO(3) matrix, can be expressed as a 6D vector by using two 3D vectors that correspond to the directions of the two orthogonal axes. Consequently, ours model predicts the rotation as a 6D-vector and employs the geodesic loss. In summary, let $T^*$ and $R^*$ denote the ground-truth, and for simplicity, denote

$D_{KL}\big(q(z|x, f)||\mathcal{N}(0,1)\big)$ as $D(x, f)$, and denote the concatenated features $(f_s, f_v)$ as $f_{sv}$. The loss function is:

$$\mathcal{L}_T = \mathcal{L}_1(T, T^*) + \mathcal{L}_{geo}(R, R^*) + D(T^*, f_{sv}) + D(R^*, f_{sv}).$$

For the losses used in the Bi-Affordance Predictor, we provide detailed explanation in Appendix C.

## 5. Benchmark

### 5.1. Simulation Benchmark

Constructing a large-scale dataset with real objects is both time-consuming and costly. To address this challenge, we utilize the Breaking Bad Dataset (Sellán et al., 2022), which models the natural destruction of objects into fragments. This dataset features multiple categories, diverse objects, and varying fracture patterns. For physics simulation, we employ the SAPIEN (Xiang et al., 2020) platform along with two two-finger Franka Panda grippers as robot actuators.

We randomly select a pair of 3D fractured parts from a randomly chosen shape within a random category. The initial part poses are also randomized. Given the considerable diversity in fractured parts, collecting successful manipulation data for assembly can be quite challenging. To enhance data collection efficiency, we implement several heuristic strategies, with details in Appendix B.

### 5.2. Real-World Benchmark

A real-world benchmark is crucial not only for the evaluation of various methods but also for providing a standardized platform that enables researchers to reproduce and share solutions. Illustrated in Figure 3, we build the real-world benchmark by scanning with a smart phone camera. First,

*Table 1.* Quantitative results for novel instances within training categories and for unseen categories.

| Method | Novel Instances in Training Categories | | | | | | | | | | AVG | Novel Categories | | | | | AVG |
|---|---|---|---|---|---|---|---|---|---|---|---|---|---|---|---|---|---|
| ACT | 2% | 0% | 0% | 0% | 1% | 0% | 0% | 0% | 0% | 0% | 0.30% | 0% | 1% | 0% | 0% | 1% | 0.4% |
| Heuristic | 5% | 8% | 0% | 3% | 2% | 4% | 3% | 5% | 10% | 2% | 4.20% | 1% | 5% | 2% | 0% | 14% | 4.4% |
| SE(3)-Equiv | 0% | 0% | 4% | 0% | 1% | 5% | 0% | 7% | 2% | 11% | 3.00% | 4% | 0% | 2% | 0% | 2% | 1.6% |
| DualAfford | 21% | 17% | 0% | 2% | 2% | 4% | 14% | 8% | 10% | 6% | 8.40% | 5% | 10% | 4% | 1% | 16% | 7.2% |
| w/o SE(3) | 59% | 29% | **13%** | **14%** | 11% | 8% | 15% | **19%** | 24% | 20% | 21.20% | 13% | 24% | 4% | **9%** | 22% | 14.4% |
| Ours | **60%** | **38%** | **13%** | 13% | **12%** | **9%** | **26%** | 18% | **27%** | **25%** | **24.10%** | **14%** | **31%** | **10%** | 7% | **25%** | **17.4%** |
| w/ GT Target | 71% | 28% | 4% | 9% | 9% | 13% | 27% | 19% | 25% | 19% | 22.40% | 14% | 33% | 12% | 9% | 27% | 19% |

we put the object on an automatic turntable with 6 aruco markers around for precise camera localization, and capture a RGB video from a top-down view to a level view, lowering the height by one level for each 360-degree rotation. After capturing 4-5 levels, we uniformly sample around 300 frames, and feed them to COLMAP (Schönberger & Frahm, 2016; Schönberger et al., 2016) for estimating camera poses. Then, we use Grounded SAM 2 (Ren et al., 2024a; Ravi et al., 2024) to generate object masks and Depth Anything V2 (Yang et al., 2024) to predict monocular depths, and use SDFStudio (Yu et al., 2022; Wang et al., 2021) with depth ranking loss (Wang et al., 2023) to reconstruct object mesh. To annotate the ground-truth of scanned object assembly, we import the object slices to Blender (Community, 2018) and edit the object transformations.

Our real-world benchmark encompasses a diverse range of object categories, including wine glass, plate, beer bottle, bowl, mug, and teapot. These objects have been primarily selected from well-known international brands, ensuring both durability and accessibility. To promote object diversity, our shapes vary in size, geometry, transparency, and texture, with different seam geometries.

# 6. Experiments

## 6.1. Simulation and Settings

The simulation environment is built on the SAPIEN (Xiang et al., 2020) platform, utilizing the Franka Panda grippers as the robot actuator. We employ the EverydayColorPieces dataset from the Breaking Bad Dataset (Sellán et al., 2022), covering 15 categories, 445 shapes and 11,820 fragment pairs, with 10 categories for training and the remaining 5 for testing. Training categories are further divided into training shapes and novel instances, allowing the evaluation on generalization capabilities at both the object and category levels. More details can be found in Appendix A. For each method, we provide 7,000 positive and 7,000 negative samples. Negative samples encompass failures occurring during grasping, alignment, and assembly steps.

## 6.2. Evaluation Metric, Baselines and Ablation

**Evaluation Metric.** Our metric evaluates whether relative distance (measured in unit-length) and rotation angle (measured in degrees) of two parts are within the threshold range at the end of assembly. These thresholds ensure the success of assembly can be measured consistently and meaningfully. To evaluate each method, we prepare 100 samples in each category. For each sample, all methods are presented with the same initial observation for a fair comparison.

**Baselines and Ablations.** We compare with four baselines and two ablated versions:

(1) ACT (Zhao et al., 2023), a transformer network with action chunking that imitates successful action sequences in closed-loop manner. We enhance this method by providing depth information, object pose, and an additional target image as inputs. Besides, this baseline is trained and tested on individual categories, whereas other learning-based methods are trained on all training categories and evaluated on both novel instances and unseen categories.

(2) Heuristic, where we hand-engineer a set of heuristic strategies to improve manipulation success rate. These strategies are similar to the data collection heuristics (Appendix B).

(3) SE(3)-Equiv (Wu et al., 2023c) that learns SE(3)-equivariant part representations to estimate target assembled part poses in the computer vision domain. Though originally designed for assembled part pose predictions without considering robotic execution, we adapted it as a baseline by generating the pick-up actions of robots via heuristic strategies. Additionally, given the predicted pose $q_i^{asm}$ for part $i$, we compute the corresponding gripper pose $g_i^{asm}$ for assembly action, following Equation 1.

(4) DualAfford (Zhao et al., 2022) that learns collaborative visual affordance for bimanual manipulation. While it focuses on short-term manipulation, we adapt it to determine where to grasp the two fractured parts, using heuristic methods for alignment and assembly steps.

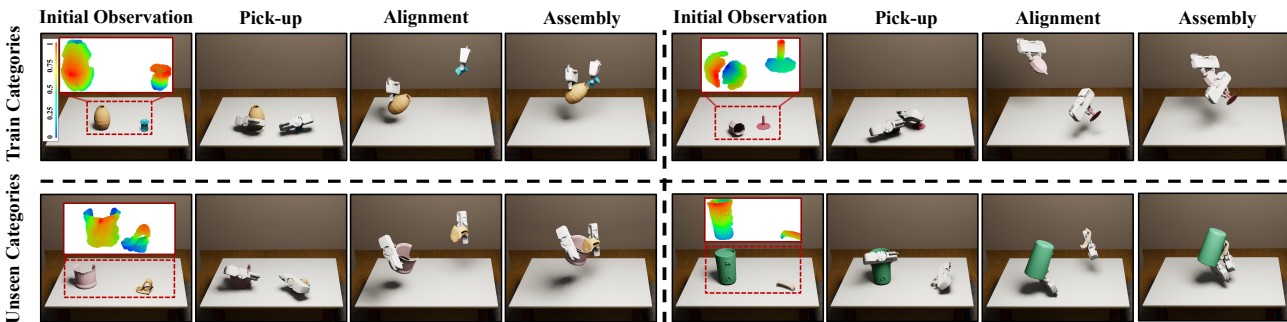

*Figure 4.* **Simulation Experiment.** We present qualitative results of the predicted affordance maps and robot actions from our method.

(5) w/o SE(3), an ablation that replaces SO(3)-equivariant VN-DGCNN encoder with PointNet++.

(6) w/ GT target, where we provide additional ground-truth disassembly direction $v$ and transformation $M$. Ground truth is sampled using a heuristic method that ensures at least one feasible assembly.

## 6.3. Quantitative Results and Analysis

Table 1 shows the success rate comparisons across different methods on both the novel instance dataset and the unseen category dataset. Our method outperforms the baselines and ablation models in most cases, demonstrating its effectiveness and geometric generalization capabilities.

For **ACT**, though we provide additional input such as depth, object poses, and the goal image, it achieves lower scores on our task. Although ACT successfully picks up parts in approximately 40% of trials, it often fails during the alignment phase, with a misalignment of over 100° between the parts in many cases. Furthermore, ACT struggles to avoid grasping the fractured seam regions, leading to collisions during the assembly process. This is because the observation and action spaces in robotic geometric assembly are exceptionally large, making it challenging to directly learn the appropriate fine-grained actions.

For **heuristic**, it achieves higher scores because we provide substantial ground-truth information in simulation. However, due to the significant diversity in both inter- and intra-category shape geometries, it is unrealistic to expect hand-engineered rules to generalize effectively across all shapes.

For **SE(3)-Equiv**, the lower accuracy is primarily due to its disregard for the robotic execution process, which is a common limitation of prior visual assembly estimation methods. Specifically, these methods do not determine suitable grasping locations on the fragments, not only for successful pick-up but also for avoiding the seam region critical to subsequent assembly. They also lack the capability to align the fragments properly at the seam, significantly increasing the likelihood of collisions when the parts are brought together.

For **DualAfford**, it performs better than the heuristic policy, demonstrating its superior ability to learn geometric-aware pick-up poses compared to heuristic sampling. However, only with designs focused on short-term manipulation, it still lacks awareness of the subsequent alignment and assembly steps.

In comparison with the ablation models, our method also shows consistent improvements. Compared to the ablation **w/o SE(3)**, we observe that utilizing the SE(3)-equivariant representation enhances the performance across most categories. Compared to the ablation **w/ GT Target**, our method performs better on novel instances but worse on novel categories. This suggests that, for the training categories, our method learns to predict a more accurate distribution of disassembly and transformation, surpassing those sampled from the heuristic strategy. However, on novel unseen categories, while our method still demonstrates generalization capability, the ablated version with the ground-truth target remains more effective.

Appendix E presents additional experiments and analyses, including multi-fragment assembly, the use of "imperfect" imaginary assembly shape inputs, and further ablations.

## 6.4. Qualitative Results and Analysis

In Figure 4, we present collaborative affordance maps and robot manipulations predicted by our methods across multiple categories, including novel instances in training categories and unseen categories. Predicted affordance demonstrates an awareness of part geometry, highlighting graspable regions while avoiding areas near the table that could result in collisions between the gripper and surface. Additionally, the affordance accounts for subsequent alignment and assembly steps, avoiding seam areas that may cause collisions during the approach phase. Based on predicted affordance map, our model predicts appropriate gripper actions for assembling parts. Moreover, the results demonstrate the ability to generalize to unseen categories and shapes.

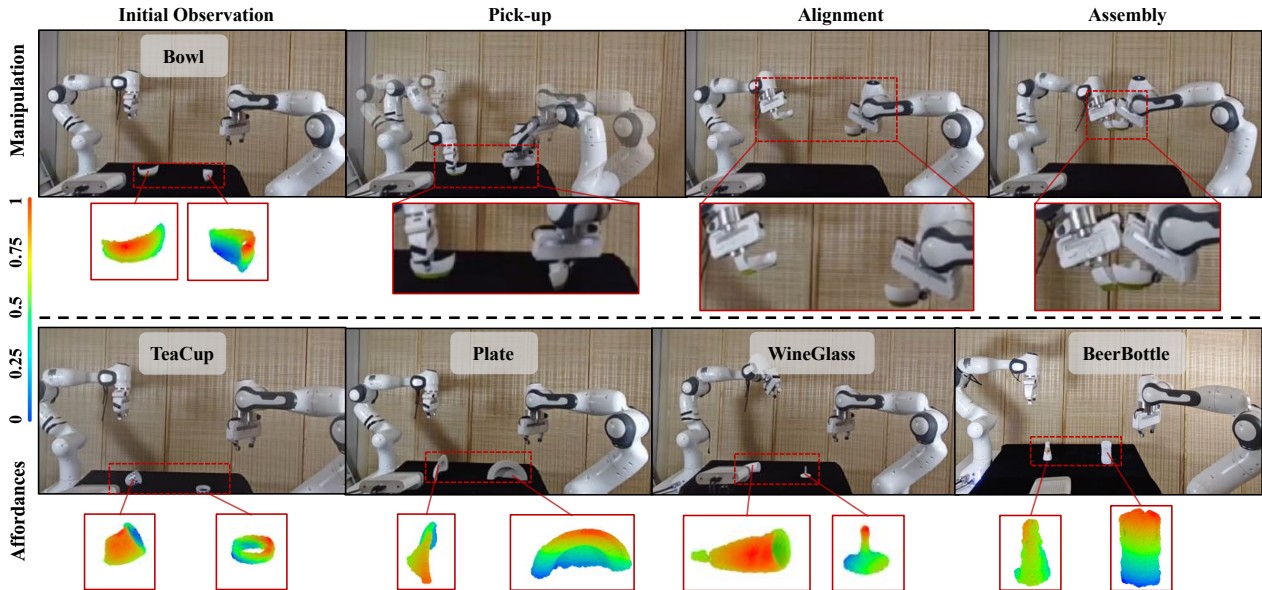

*Figure 5.* **Real-World Experiment.** We present the results of our model tested on real-world scans.

### 6.5. Real-World Experiments

We set up two Franka Panda with fractures positioned between them. An Azure Kinect camera is mounted in front of the robots, capturing partial 3D point cloud data as inputs for our models. The robots are controlled via the Robot Operating System (ROS) (Quigley et al., 2009), with control and communication managed through the frankapy library (Zhang et al., 2020). Communication with the Kinect Azure is facilitated by the pyk4a library (pyk4a, 2019).

Figure 5 and the bottom row of Figure 1 present promising results by directly testing our method in real-world scenarios. Detailed manipulation processes are shown in Appendix F and videos are available in the supplementary materials. We observe that our model not only learns which regions of the fractured parts to grasp but also avoids manipulating areas near the fracture regions or too close to the table surface, reducing the likelihood of collisions during manipulation. The results from the real-world experiments demonstrate our model's capacity for generalization to real-world scenarios.

### 7. Conclusion

We leverage the geometric generalization capability of point-level affordance to develop a method that enables both generalization and collaboration in long-horizon geometric assembly tasks. To evaluate performance across diverse geometries, we introduced a real-world benchmark that features significant geometric variety and global reproducibility. Extensive experiments have shown that our approach outperforms previous methods, demonstrating its effectiveness in handling complex and long-horizon assembly tasks.

### Impact Statement

This work contributes to the development of shape assembly skills in future robots, specifically focusing on visual collaborative affordance predictions. These affordances can be visualized and analyzed prior to deployment, which may reduce potential risks and dangers. We do not anticipate any significant ethical concerns or societal consequences arising from our research.

### Acknowledgements

This paper is supported by National Natural Science Foundation of China (No. 62376006) and National Youth Talent Support Program (8200800081).

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

# Appendix

## A. Details about Data Statistics

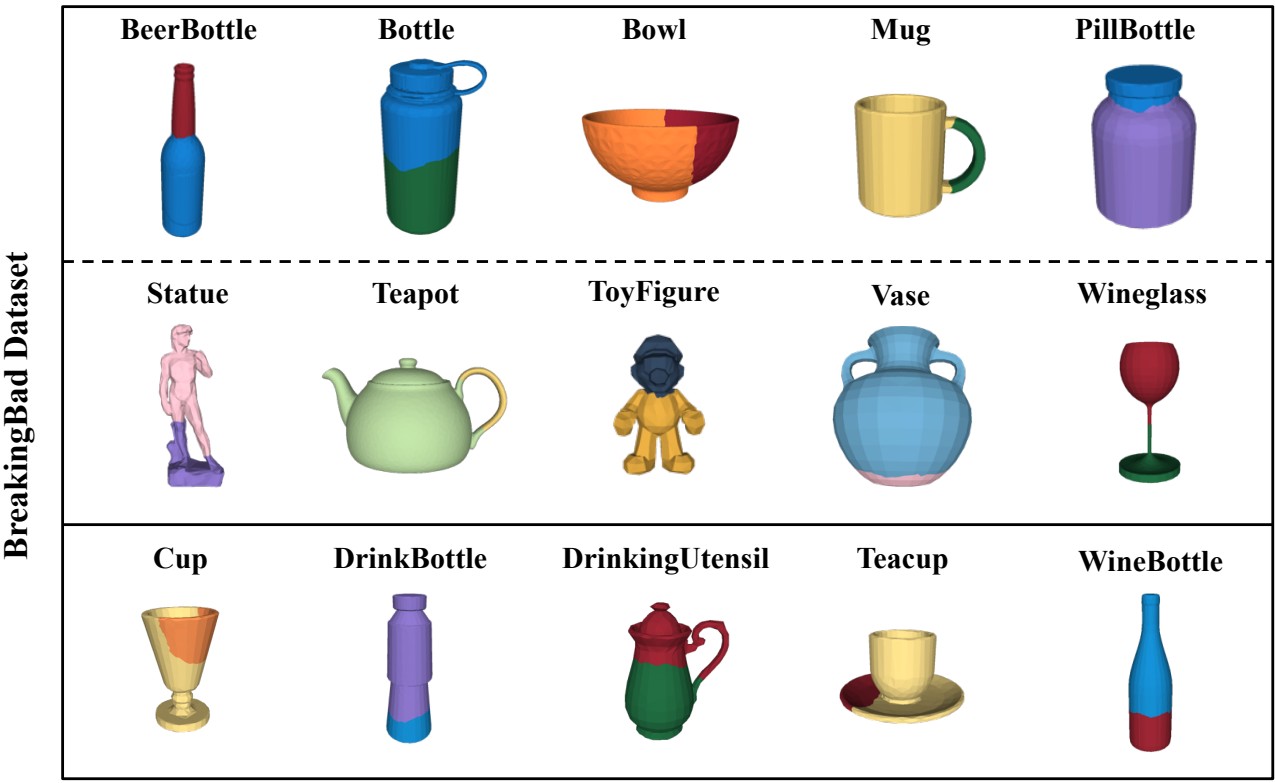

*Figure 6.* Visualization of simulation data. We present one example shape from each object category used in our paper.

In Figure 6, we present a representative example for each object category from the dataset used in our experiments.

In this paragraph, we detail the data split for our experiments. We randomly select 10 out of the 15 categories for training, reserving the remaining 5 categories exclusively for testing. Within the 10 training categories, 60% of the shapes are randomly chosen for the training set, while the remaining 40% serve as a test set to assess the models' performance on novel instances within the training categories (shape-level). For the reserved 5 categories, all shapes are included in the test set to evaluate the methods' generalization capabilities on unseen categories (category-level). In summary, the training set consists of 10 categories, totaling 237 shapes and 6,403 pairs of fragments. The shape-level test set includes 10 categories, comprising 131 shapes and 3,638 pairs of fragments. The category-level test set encompasses 5 categories, containing 77 shapes and 1,779 pairs of fragments. Detailed statistics for each category can be found in Table 2.

## B. Details about Data Collection in Simulation

In this section, we provide detailed information about data collection in the simulation.

Due to the complexity of bimanual geometric assembly tasks, which stems from the vast observation and action spaces, it is nearly impossible to directly acquire positive data by randomly manipulating the fractured parts. To address this, we apply several heuristic strategies to improve the efficiency of data collection. Specifically, our strategies focus on the following three key steps in the process:

1. Sampling the grasping poses for the two grippers

2. Sampling the alignment poses for the two grippers

3. Sampling the assembly poses for the two grippers

*Table 2.* Shape and Fracture Counts Across Categories. Numbers before the slash represent the training set, and numbers after the slash represent the testing set. The top 10 categories are the training categories, and the bottom 5 categories are the unseen categories.

| Category | Shape (Train/Test) | Fracture (Train/Test) |
|---|---|---|
| BeerBottle | 6 / 3 | 100 / 61 |
| Bottle | 51 / 22 | 1296 / 559 |
| Bowl | 16 / 32 | 446 / 801 |
| Mug | 32 / 15 | 876 / 545 |
| PillBottle | 7 / 3 | 217 / 60 |
| Statue | 2 / 0 | 57 / 35 |
| Teapot | 7 / 3 | 315 / 104 |
| ToyFigure | 36 / 16 | 1118 / 556 |
| Vase | 74 / 32 | 1842 / 872 |
| WineGlass | 6 / 3 | 136 / 45 |
| Cup | 0 / 31 | 0 / 663 |
| DrinkBottle | 0 / 7 | 0 / 230 |
| DrinkingUtensil | 0 / 14 | 0 / 343 |
| Teacup | 0 / 7 | 0 / 167 |
| WineBottle | 0 / 18 | 0 / 376 |
| Total | 237 / 208 | 6403 / 5417 |

Each of these steps is described in detail in the following subsections.

### B.1. Sampling Grasping Poses

Different from furniture assembly task, where grasp points are easier to define, the objects in geometric shape assembly tasks have more diverse geometries, making it challenging to establish a consistent grasping policy. As a result, our heuristic strategy for grasping primarily focuses on the orientation of the grippers rather than specific grasp points.

At initialization, the two parts are randomly placed on the table. From the simulation, we obtain the ground-truth depth map and normalization map of the two parts. The normal directions often closely align with feasible grasping directions (i.e., the z-axis of the gripper). Consequently, we randomly select a grasp point on the part, and then choose a grasping direction within a cone that deviates less than 30 degrees from the normal direction at that point. To avoid potential collisions between the grippers and the table during grasping, we discard any directions that point towards the upper hemisphere of the world coordinate system.

In addition to the gripper's z-axis, the x-axis also significantly impacts grasping accuracy. Therefore, we uniformly sample a list of $n$ x-axis candidates that are orthogonal to the gripper's z-axis. By combining each candidate x-axis with the z-axis, we determine the gripper pose. We test each of these gripper poses sequentially. If a grasp pose successfully grasps the object, we proceed to the next stage; otherwise, we reset the scene and move on to the next x-axis candidate. If all grasp pose candidates fail, we record this as negative data for the grasping step. In our implementation, we empirically set $n = 6$, resulting in each x-axis candidate being spaced 60 degrees apart.

### B.2. Sampling Alignment Poses

To sample the grippers' alignment poses, we begin by sampling feasible part poses during the alignment step. Our heuristic strategy also follows a reverse disassembly process. Specifically, we load the ground-truth assembled object into the simulation at a height of 0.5 meters above the tabletop, and allow the assembled object to take any pose rather than being restricted to a canonical pose. Next, we randomly explore feasible disassembly directions for the two parts, ensuring that these directions are collision-free. The resulting poses of the parts, after moving in their respective disassembly directions, represent the parts' alignment poses. It is important to note that we will discard alignment poses that are too distant from the parts' initial poses (for example, if the initial left part has an alignment pose to the right, while the initial right part has an alignment pose to the left).

Once we have determined the parts' alignment poses, we can calculate the grippers' alignment poses using the functions described in Section 4.5.

### B.3. Sampling Assembly Directions

In the previous step, we identified the feasible disassembly directions for the ground-truth assembled parts. Consequently, we can obtain the assembly directions by simply inverting these disassembly directions, allowing the two grippers to assemble the parts accordingly. However, it is important to note that this assembly process may lead to failures. This is because, although the parts can be successfully aligned in an idealized scenario without grippers, the presence of grippers increases the risk of collisions. For instance, if one gripper is positioned too close to the seam area of a fractured part, it may collide with another part or the other gripper during the assembly process.

## C. More Details about the BiAffordance Framework

In this section, we provide more details about the BiAffordance Predictor. Following DualAfford (Zhao et al., 2022), we decompose the bimanual cooperation task into two separate yet closely interconnected submodules, $\mathcal{M}_1$ and $\mathcal{M}_2$, which conditionally predict the first and second gripper actions, respectively.

During inference, the first module $\mathcal{M}_1$ predicts the first gripper action $g_1 = (p_1^*, r_1)$, followed by the second module $\mathcal{M}_2$, which predicts the second action $g_2 = (p_2^*, r_2)$ conditioned on $g_1$, as described in Section 4.4 of the main paper.

During training, $\mathcal{M}_2$ still takes the first gripper action $g_1$ as input, and then generates a complementary second action $g_2$. However, since $\mathcal{M}_1$ lacks knowledge of how $g_2$ will be predicted, it faces challenges in predicting a collaborative action $g_1$. To address this issue, we aim to make $\mathcal{M}_1$ aware of the types of actions that can be easily collaborated on. We assess the quality of $\mathcal{M}_1$'s actions by evaluating whether $\mathcal{M}_2$ can generate cooperative actions, which encourages $\mathcal{M}_1$ to predict actions with high collaborative quality. Following this approach, $\mathcal{M}_2$ guides the training of $\mathcal{M}_1$. Thus, we first train $\mathcal{M}_2$ and then use the trained $\mathcal{M}_2$ to train $\mathcal{M}_1$, ensuring cooperative predictions.

During training, each submodule $\mathcal{M}_i$ consists of three components: (1) an Affordance Network $\mathcal{A}_i$, which predicts an affordance map to indicate where interaction should occur; (2) an Actor Network $\mathcal{U}_i$, which predicts manipulation orientations to determine how to interact at the selected point; and (3) a Critic Network $\mathcal{C}_i$, which assesses the likelihood of the action's success.

To explain the training process, we begin with the more straightforward second module, $\mathcal{M}_2$, which is also the first to be trained.

The Actor Network $\mathcal{U}_2$ in $\mathcal{M}_2$ is implemented as a conditional Variational Autoencoder (cVAE). As detailed in Section 4.4, it takes concatenated input features $f_{in} = (f_p, f_{S'}, f_{v'})$ and the ground-truth feature of the first action $f_{g_1} = (f_{p_1^*}, f_{r_1})$ from the collected data. We apply a geodesic distance loss to measure the error between the reconstructed gripper orientation $r_2$ and the ground-truth orientation $\hat{r}_2$, along with KL divergence to quantify the difference between the two distributions:

$$\mathcal{L}_{\mathcal{U}_2} = \mathcal{L}_{geo}(r_2, \hat{r}_2) + D_{KL}\big(q(z|\hat{r}_2, f_{in}, f_{g_1})||\mathcal{N}(0, 1)\big). \tag{4}$$

The Critic Network $\mathcal{C}_2$ in $\mathcal{M}_2$ is implemented as a multilayer perceptron (MLP) and evaluates how well the predicted second gripper action $g_2 = (p_2^*, r_2)$ collaborates with the first action $g_1$. Using the collected data along with the corresponding ground-truth interaction results $r$ (where $r = 1$ indicates a positive interaction and $r = 0$ indicates a negative one), we train $\mathcal{C}_2$ with the standard binary cross-entropy loss:

$$\mathcal{L}_{\mathcal{C}_2} = r_j \log\big(\mathcal{C}_2(f_{in}, f_{g_1}, f_{p_2^*}, f_{r_2})\big) + (1 - r_j)\log\big(1 - \mathcal{C}_2(f_{in}, f_{g_1}, f_{p_2^*}, f_{r_2})\big). \tag{5}$$

The Affordance Network $\mathcal{A}_2$ in $\mathcal{M}_2$ is implemented as a multilayer perceptron (MLP). The predicted affordance score represents the expected success rate for executing action proposals generated by the Actor Network, which can be directly evaluated by the Critic Network. To obtain the ground-truth affordance score $\hat{a}_{p_i}$ on $p_i$, we use the Actor Network $\mathcal{U}_2$ to

sample $n$ gripper orientations at the point $p_i$ and calculate the average action scores assigned by the Critic Network $\mathcal{C}_2$. We apply L1 loss to measure the error between the predicted and ground-truth affordance scores at a specific point $p_i$:

$$\hat{a}_{p_i} = \frac{1}{n} \sum_{j=1}^{n} \mathcal{C}_2\big(f_{in}, f_{g_1}, f_{p_i}, \mathcal{U}_2(f_{in}, f_{g_1}, f_{p_i}, z_j)\big); \quad \mathcal{L}_{\mathcal{A}_2} = |\mathcal{A}_2(f_{in}, f_{g_1}, f_{p_i}) - \hat{a}_{p_i}|. \tag{6}$$

After training the expert model $\mathcal{M}_2 = (\mathcal{A}_2, \mathcal{U}_2, \mathcal{C}_2)$, we can utilize it to generate collaborative actions $g_2$ for a given $g_1$ predicted by $\mathcal{M}_1$. Thus, we can assess the quality of $\mathcal{M}_1$'s actions by evaluating whether $\mathcal{M}_2$ can generate cooperative actions. Specifically, to evaluate the predicted $g_1$, we use the trained $\mathcal{A}_2$ and $\mathcal{U}_2$ to generate multiple second gripper action candidates $\{g_2\}$. We then employ $\mathcal{C}_2$ to determine how well these second gripper candidates $\{g_2\}$ collaborate with the proposed $g_1$. The average critic score reflects how easily the second gripper can cooperate with the proposed first action $g_1$. Consequently, this average score serves as the ground truth for the first Critic Network $\mathcal{C}_1$, and we apply L1 loss for supervision:

$$\hat{c}_{g_1} = \frac{1}{nm} \sum_{j=1}^{n} \sum_{k=1}^{m} \mathcal{C}_2\big(f_{in}, f_{g_1}, f_{p_j}, \mathcal{U}_2(f_{in}, f_{g_1}, f_{p_j}, z_{jk})\big); \quad \mathcal{L}_{\mathcal{C}_1} = |\mathcal{C}_1(f_{in}, f_{g_1}) - \hat{c}_{g_1}|. \tag{7}$$

To train the Affordance Network $\mathcal{A}_1$ and Actor Network $\mathcal{U}_1$ in $\mathcal{M}_1$, the loss functions are similar to those used for $\mathcal{A}_2$ and $\mathcal{U}_2$. Therefore, with the trained Critic Network $\mathcal{C}_1$, the Affordance Network $\mathcal{A}_1$ assigns high scores to points that can be easily manipulated collaboratively by the subsequent gripper action.

In this training pipeline, the two gripper modules can generate collaborative affordance maps and manipulation actions for bimanual tasks. Note that during inference, the use of the Critic Networks is optional.

## D. Details about Training and Computational Costs

During training, there are two main components: (1) the Disassembly Predictor and the Transformation Predictor are trained together in an end-to-end manner, and (2) all modules within the BiAssembly Predictor are also trained collectively in an end-to-end manner. These two training components can be conducted simultaneously on a single GPU. Using a single NVIDIA V100 GPU, the total training time for our model is approximately 48 hours: the combination of the Disassembly Predictor and Transformation Predictor converges in about 20 hours, while the BiAffordance Predictor converges in about 48 hours.

During inference, our method utilizes only 1,600 MB of GPU memory and processes each data point in an average of 0.1 seconds.

## E. More Experimental Results

### E.1. Handling Multiple Broken Parts

*Table 3.* Quantitative Results for multi-fragment assembly.

| Method | | | | | | AVG |
|---|---|---|---|---|---|---|
| Ours | 24% | 22% | 16% | 12% | 9% | 16.6% |

Our method can be extended to handle multi-fragment assembly tasks. In Table 3, we present the accuracy of our method on three-fragment assembly tasks. Additionally, Figure 7 showcases visualizations that include the predicted affordance maps and collaborative actions. Both the quantitative and qualitative results demonstrate that our proposed method can be

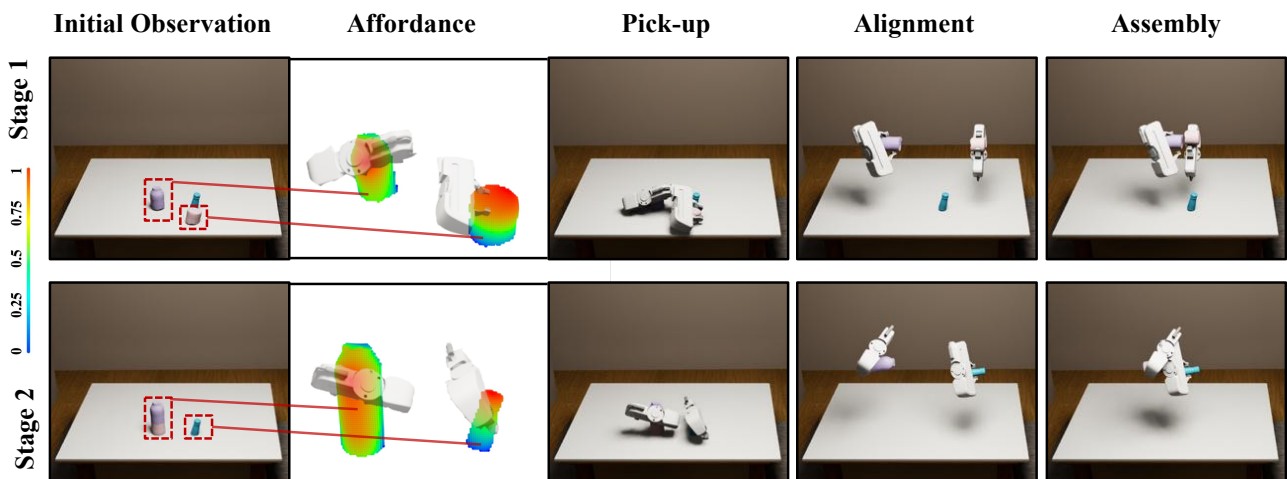

*Figure 7.* We provide the visualization of the predicted affordance maps and actions for multi-part assembly task.

effectively adapted to multi-fragment assembly tasks. Below, we provide a detailed explanation of how our method can be adapted for multi-fragment assembly.

The multi-fragment assembly task can be achieved by iteratively applying the two-fragment assembly process. First, at each iteration, we can identify which two fragments, $p_i$ and $p_j$, should be assembled next. (If some parts have already been assembled in previous iterations, their combination is treated as a new fragment.) Specifically, based on the imaginary assembled shape $S$, we can calculate the minimum distance, $\min \|p_i - p_j\|$, between sampled points from every pair of fragments, and the pair $(p_i, p_j)$ with the minimum distance is chosen for assembly: $(p_i, p_j) = \underset{(p_i, p_j) \in \mathcal{S}_1 \times \mathcal{S}_2}{\arg\min} \|p_i - p_j\|$. Once $p_i$ and $p_j$ are identified on $S$, we then map these fragments to their corresponding parts in the observed point cloud $O$. This mapping is formulated as a classification task, where the similarity between parts in $S$ and $O$ is estimated.

Finally, using the imaginary assembled shape of the selected fragments, $S_{p_i} \cup S_{p_j}$, and the corresponding observed point cloud $O_{p_i} \cup O_{p_j}$, our method predicts the actions to pick up and assemble the fragments. This process mirrors the steps of the standard two-fragment assembly method. By iteratively applying this two-fragment assembly process, the complete assembly of all fragments can be achieved.

To validate the feasibility of this multi-fragment assembly process, we fine-tune our pretrained BiAssembly model on multi-category three-piece assembly data and present the results on unseen objects. Based on the accuracy reported in Table 3 and the visualizations in Figure 7, we can see that even for multi-fragment assembly task, our method can still generate reasonable results.

### E.2. Utilizing the Imperfect Imaginary Assembled Shape Prediction

Predicting the imaginary assembled shape from multiple fractured parts is a relatively well-studied vision problem (Sellán et al., 2022; Wu et al., 2023c; Lu et al., 2024c; Tsesmelis et al., 2024; Scarpellini et al., 2024). Previous works have demonstrated the ability to predict precise fragment poses that allow for an imaginary assembled shape, making it reasonable to assume the existence of such shapes in our framework. Additionally, in traditional furniture assembly tasks, several studies (Wang et al., 2022a; Sera et al., 2021; Wan et al., 2024) also assume the existence of an imaginary Therefore, for simplicity, we adopt the concept of the imaginary assembled shape, aligning with advancements in prior works.

While this simplification is reasonable, we also conduct experiments to evaluate our BiAssembly Method on "imperfect" imaginary assembled shapes. Specifically, we first utilize the pretrained model from a prior visual assembly method (Wu et al., 2023c) to generate the assembled point cloud prediction. We then replace the input "perfect" imaginary assembled shapes with these "imperfect" predictions. Importantly, the only difference is the change in the input imaginary shape, the subsequent BiAssembly framework remains unchanged, and we do not fine-tune the model on the new predictions. The average accuracy based on imperfect imaginary shapes is 20.80% on training categories and 17.20% on novel unseen categories, which is comparable to the accuracy on perfect imaginary shapes (24.10% on training categories and 17.40% on

novel unseen categories). These results demonstrate that our method is capable of tolerating imperfect input shapes, even without any fine-tuning process, supporting the reasonableness of our assumption for simplification.

### E.3. More Ablation Studies

In this section, we conduct three additional ablation studies, and provide the quantitative results in Table 4 for novel objects within training categories and in Table 5 for novel unseen categories.

**w/o Affordance Network:**   During inference, we do not use the trained Affordance Network to highlight actionable regions. Instead, we randomly sample a contact point on the part. The results show a significant drop in the success rates, which decrease to 4.60% for training categories and 2.80% in unseen categories. This demonstrates that the Affordance Network plays a crucial role in filtering out non-graspable points and points that are unsuitable for the subsequent assembly process.

**w/o Transformation Predictor:**   In this ablation, we remove the Transformation Predictor during inference. This results in success rates of 7.40% on training categories and 4.80% on unseen categories, both substantially lower than our original method. These results show that the Transformation Predictor plays an essential role in predicting alignment poses, enabling the robot to manipulate parts from their initial to alignment poses without collisions.

**w/ heuristic disassembly direction $v$:**   In this case, we remove the Disassembly Predictor during inference. Instead, we compute the center of each part from the imaginary assembled shape $S$ by averaging the part points, and then use the relative direction of the two parts' centers as the disassembly direction $v$. This ablation achieves success rates of 19.70% on training categories and 15.20% on unseen categories, both of which are lower than those achieved by our method. While this ablated version performs well on certain categories, suggesting that the calculated relative direction can approximate the relative positions of the two parts, it falls short in categories with complex geometries. In such cases, the heuristic method lacks the accuracy needed to replace the assembly direction required for our task. This highlights the critical role of the Disassembly Predictor in achieving better performance.

*Table 4.* More ablation studies: quantitative results in novel instances within training categories.

| Method | Novel Instances in Training Categories | | | | | | | | | | AVG |
|---|---|---|---|---|---|---|---|---|---|---|---|
| w/o Affordance | 7% | 11% | 0% | 0% | 1% | 8% | 1% | 4% | 6% | 8% | 4.60% |
| w/o Transformation | 29% | 19% | 0% | 0% | 0% | 0% | 8% | 4% | 5% | 9% | 7.40% |
| w/ heuristic $v$ | 54% | 28% | 0% | 3% | 10% | 5% | **28%** | **23%** | 21% | **25%** | 19.70% |
| Ours | **60%** | **38%** | **13%** | **13%** | **12%** | **9%** | 26% | 18% | **27%** | **25%** | **24.10%** |

*Table 5.* More Ablation studies: quantitative results in the novel unseen categories.

| Method | Unseen Categories | | | | | AVG |
|---|---|---|---|---|---|---|
| w/o Affordance | 2% | 6% | 2% | 0% | 4% | 2.8% |
| w/o Transformation | 4% | 10% | 1% | 0% | 9% | 4.8% |
| w/ heuristic $v$ | **18%** | 22% | **15%** | **9%** | 12% | 15.20% |
| Ours | 14% | **31%** | 10% | 7% | **25%** | **17.4%** |

### E.4. More Bimanual Tasks

While we demonstrate the effectiveness on the challenging geometric shape assembly task, our approach is also applicable to a broader range of bimanual tasks that require coordinated manipulation. To illustrate this generalizability, we present an additional experiment on a bottle cap closing task. This task can be formulated into three sequential steps: (1) The two arms pick up the bottle and cap from the table. (2) The two arms align the cap with the bottle opening. (3) The two arms place

and secure the cap onto the bottle. Our results show that, after training on a few bottle shapes, our method generalizes well to novel bottle shapes, achieving an average accuracy of 67%. Figure 8 provides visualizations of the predicted affordance maps and the manipulation process.

| **Initial Observation** | **Affordance** | **Pick-up** | **Alignment** | **Assembly** |
| --- | --- | --- | --- | --- |

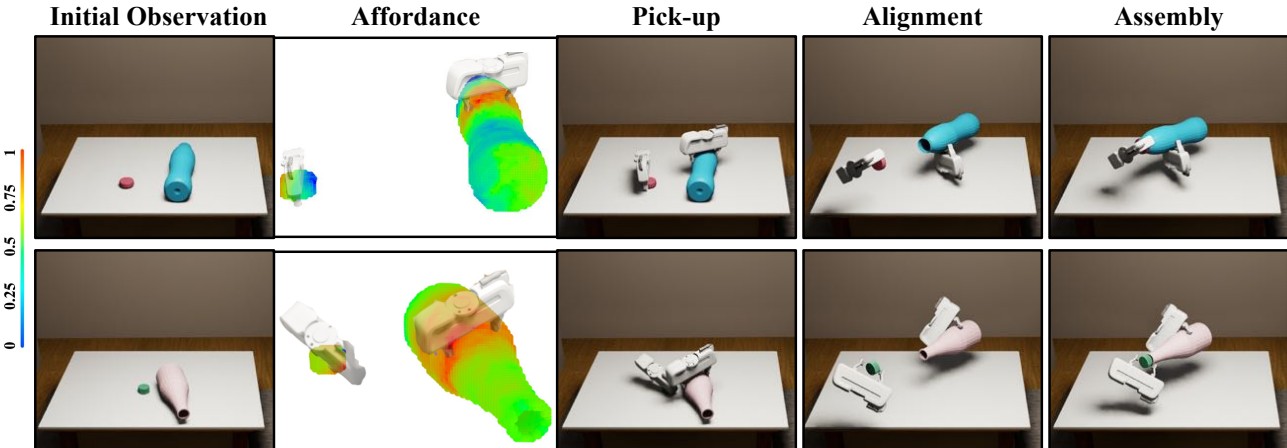

*Figure 8.* We provide visualizations of the predicted affordance maps and the manipulation process for the bottle cap closing task.

## F. More Visualizations

In Figure 9, we present a detailed visualization of the manipulation process in real-world experiments, serving as a supplement to Figure 5 in the main paper.

In Figure 10, we present additional qualitative results from our method.

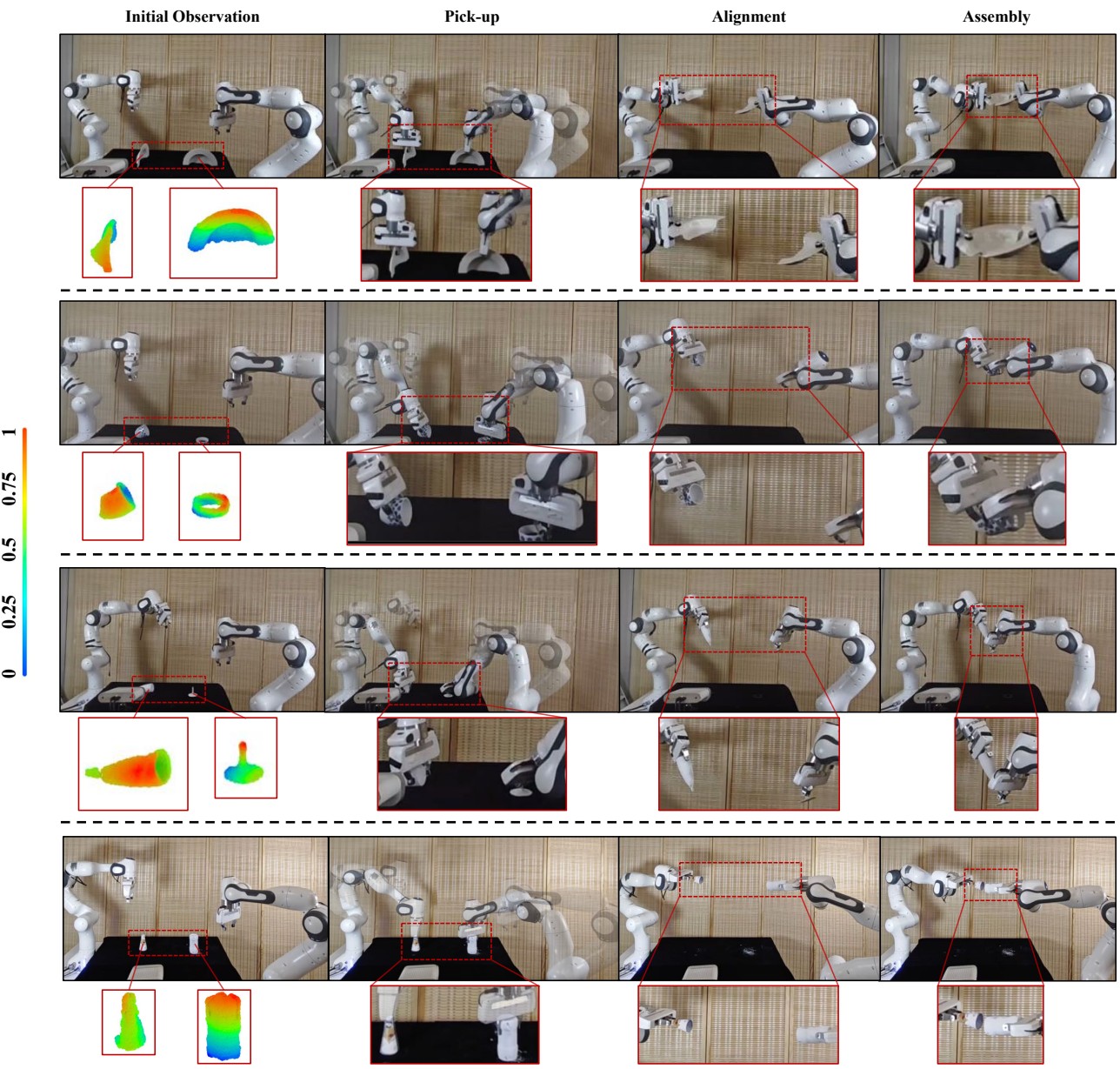

*Figure 9.* **Real-World Experiment.** We present the results of our model tested on real-world scans. For each data, We visualized the affordance map, and the bimanual actions for the pick-up, alignment, and assembly steps. Manipulation videos can be found in our supplementary materials.

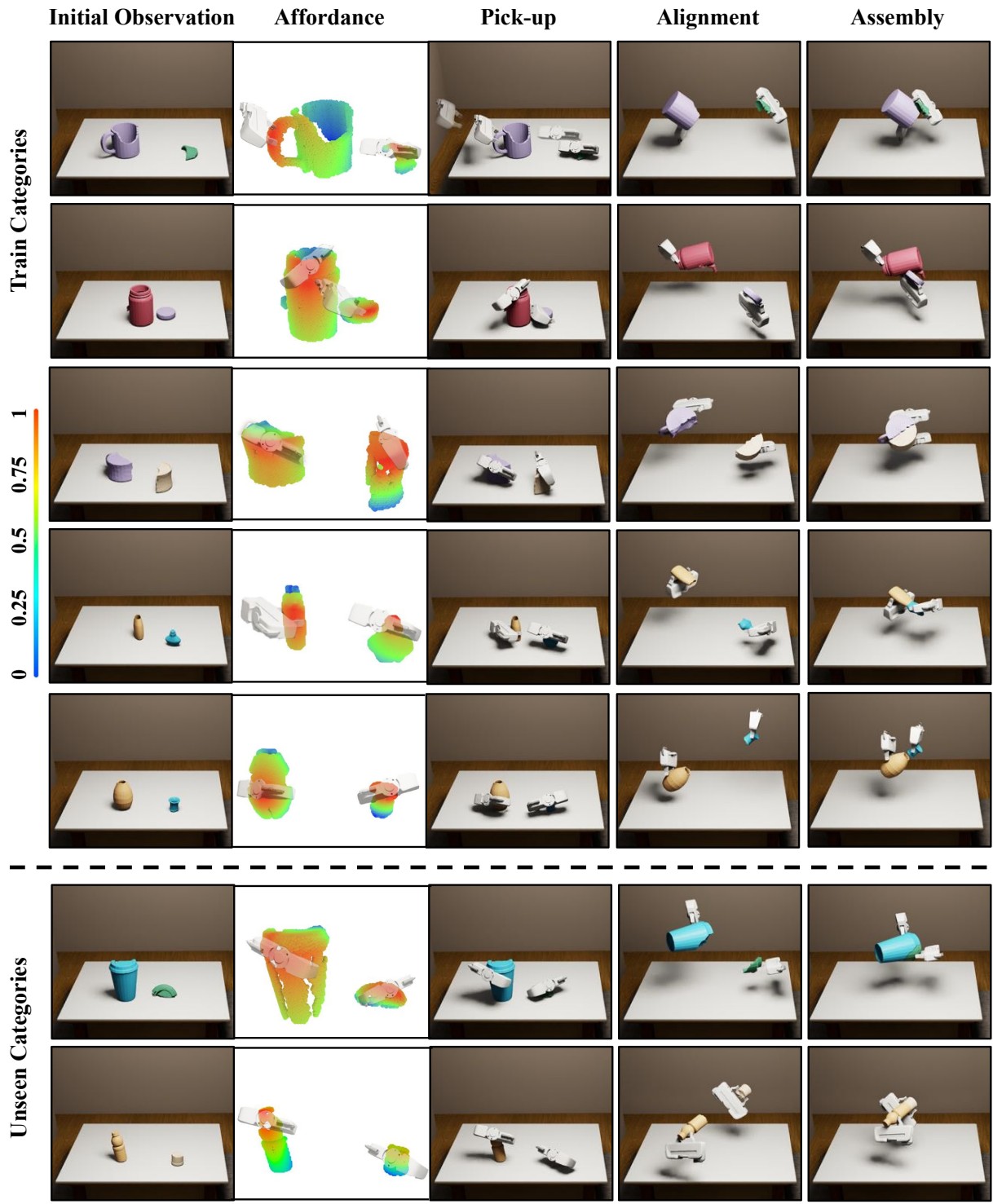

*Figure 10.* We visualize additional qualitative results that augment Figure 4 in the main paper. In each row, from left to right, we respectively present the input observation, the predicted affordance maps for the two fractured parts, and the bimanual actions for the pick-up, alignment, and assembly steps. In the top part are novel shapes from the training categories, while in the bottom part are shapes from unseen categories.

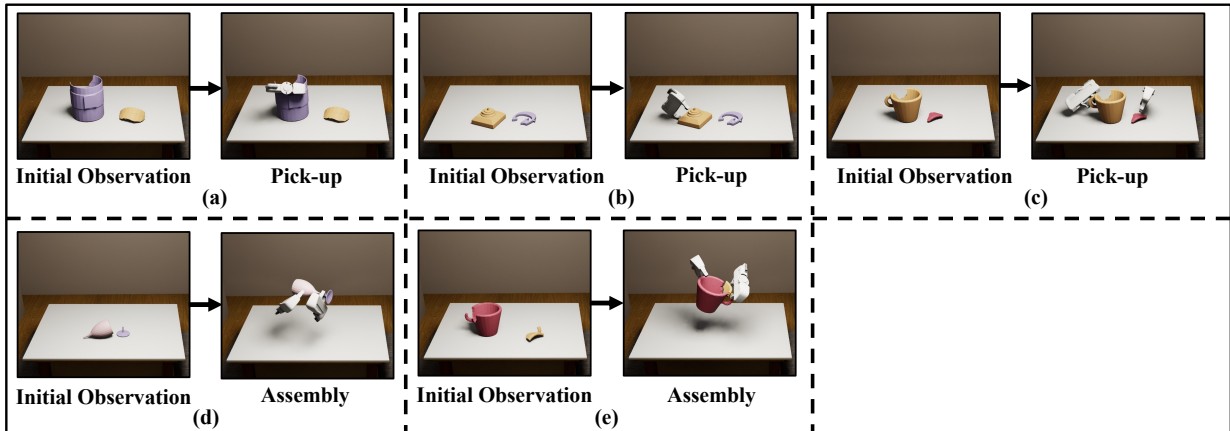

*Figure 11.* We visualize some failure cases, which demonstrate the challenges of the tasks and some cases that are difficult for robots to to determine appropriate actions. The first row presents three cases where the fractured parts are either too large or too flat to grasp. The second row includes two cases where the graspable region corresponds exactly to the seam areas; while the objects can be grasped, collisions may occur during the assembly of the parts.

## G. Failure Cases

We provide a detailed analysis of failure cases and illustrate the inherent difficulty of the task with scenarios that are particularly challenging for robots to figure out. Additionally, we provide insights into potential future improvements to address these complexities more effectively.

### G.1. Hard to Grasp

**Heavy or Smooth-Surfaced Parts.** Fractured parts that are heavy or have smooth surfaces often result in grasping failures. For instance, as shown in Figure 11 (a), categories such as teapots and vases, which are relatively large and feature smooth curved surfaces, exhibit notably high failure rates during grasping.

**Flat Parts.** Flat fractured parts, particularly some shapes in categories like statues and mugs, are challenging to pick up due to the limited gripping area. For example, as shown in (b), the statue part on the left is too close to the desktop and has a very small thickness, which prevent the gripper from grasping it. Similarly, in (c), the fragment on the right is too flat, making it impossible for the gripper to grasp it. A potential solution is incorporating pre-grasp operations, such as moving the fractured part to the table edge, allowing the shape to hang off slightly and thus become graspable.

### G.2. Hard to Assemble

**Graspable Regions Overlapping Seam Areas.** When the graspable regions of a fractured part align with its seam areas, collisions during assembly become frequent. This issue is common in categories such as wineglasses, mugs, and bowls. For example, as shown in Figure 11 (d), the left gripper avoids collision-prone regions, but the right gripper must grasp the neck of the wine bottle. Similarly, in (e), while the left gripper avoids collisions, the right gripper ends up grasping the handle of a mug. A potential solution is to perform a series of pick-and-place operations to adjust the object's initial pose. This adjustment can reduce the overlap between the object's graspable regions and seam areas, thereby minimizing collisions during the assembly process.

**Complex Object Shapes.** Objects with intricate shapes, like those in the statues category, pose challenges due to irregular edges and complex curves. Such designs increase the difficulty of alignment and manipulation, leading to higher failure rates during assembly.

**Relative Displacement During Operations.** Relative displacement between the gripper and fractured parts often occurs due to small contact areas and insufficient support, which can cause sliding or tipping during manipulation. For example, wine bottles with narrow necks, which have unstable center of gravity, making the gripper prone to sliding during movement and leading to operational failures.

