# OpenReview forum: "BiAssemble: Learning Collaborative Affordance for Bimanual Geometric Assembly"
_ICML.cc/2025/Conference — ICML 2025 poster_

### Official Review · Reviewer_LSLY · 2025-03-09

**Overall Recommendation:** 3

**Summary:**

This paper proposes a framework for learning collaborative affordance in bimanual geometric assembly. The task is assembling fractured parts into complete objects, which is a long-horizon task requiring pick-up, alignment, and assembly. The paper tackles this task through predicting collaborative affordance and gripper actions for bimanual geometric shape assembly. A real-world benchmark for re-assembling broken parts is created. Evaluations demonstrate the effectiveness of the approach and shows generalizability to unseen object categories in both simulated and real-world environments.

## update after rebuttal
The paper addresses a novel and useful task of bimanual geometric assembly. The authors also provide additional experimental results applying the method to other tasks (e.g., a bottle opening task). However, the concerns of relatively low real-world performance and many assumptions such as imagined assembly shapes, floating grippers, relying on pose trackers, etc. remain unaddressed. I will maintain my original score.

**Claims And Evidence:**

The paper claims to provide an effective solution for bimanual geometric assembly, but the reported success rates in experiments are low (20-30%), which does not support the reliability and practicability of the approach.

**Essential References Not Discussed:**

No.

**Experimental Designs Or Analyses:**

Thorough evaluations in both simulation and real-world environments are carried out to demonstrate the effectiveness of the approach. The model is generalizable to shapes from unseen categories.

The proposed ablations validate the role of individual components, such as disassembly prediction and SE(3)-equivariant representations in the obtained performance.

For real-world experiments, only qualitative results are presented, there is a lack of quantitative results on more object shapes and comparisons to other baselines. There is also a lack of more detailed sim2real transfer analysis, for example, side-by-side assembly comparison of an exact same set of shapes in simulation and the real world.

The evaluations in simulation are carried out with floating grippers. It would be more realistic to control grippers mounted on bi-manual arms, as there could be singularity and arm-table collision issues that are not being taken into account with the floating grippers.

**Methods And Evaluation Criteria:**

The integration of collaborative affordance prediction with geometric reasoning demonstrates potential for advancing bimanual assembly tasks.

The method has many assumptions. It assumes the availability of an ideal "imaginary assembled shape". Also, the task setup mainly considers objects with two fragments, however, in reality there could be an arbitrary number of fragments with multimodal contacts involved. Three-fragment assembly task results are shown in the supplementary material, but the success rate is quite low and cannot fully validate the scalability of the approach to multi-fragment assembly.

A real-world benchmark on geometric assembly is created, which paves way for future research on this direction.

**Other Comments Or Suggestions:**

See above.

**Other Strengths And Weaknesses:**

The paper addresses an underexplored but important task in robotics and provides a novel solution to tackle it. However, it relies on many assumptions (e.g., the availability of an "imaginary assembled shape" and the restriction to mostly two-part assemblies) and has low success rates and robustness. It is also not thoroughly validated in real-world environments of its effectiveness and scalability.

The failure analysis is helpful in categorizing errors but did not provide actionable insights or detailed solutions to address the low success rates.

**Questions For Authors:**

Why is the success rate so low? Which component is the most brittle part?

How would the accuracy of the pose estimator affect the performance? If the pose estimation is a bit off due to occlusions or sensor noises in the real world, would the model be robust to it and still manage to succeed?

**Relation To Broader Scientific Literature:**

The paper addresses a useful task that has been under-explored in previous robotics works, and provides an effective approach to solve this challenging task.

**Theoretical Claims:**

The equations in the paper are correct.

---

> ### Author Rebuttal · Authors · 2025-04-01
>
> Thank you for your thoughtful review and questions. We have carefully addressed them below.
>
> > W1. Low reported success rates
>
> The relatively low scores across all models and baselines stem from the highly diverse and complex nature of geometric shape assembly task. As detailed in Appendix G, our dataset contains varied fracture patterns, including many flat or minimal parts that are extremely difficult to grasp and assembly. We intentionally chose this challenging dataset to set a benchmark for future improvements. For instance, future methods could involve pre-grasp operations like moving flat parts to the table edge to make them graspable.
>
> While we demonstrate the effectiveness of our method in this complex task, our approach is also applicable to a broader range of bimanual tasks. Due to space limitations, we kindly refer the reviewer to our response to Reviewer3 (KY79m), W3, where we provide a detailed clarification that our method can be adapted to tasks like bottle cap closing, and has good performance (67% accuracy).
>
> > W2. Assumption of imaginary assembled shape. Two-fragment task setup. Three-fragment task' success rate.
>
> For **(A) imagined assembled shape**, we kindly refer the reviewer to our response to Reviewer1 (z3UB), W1, for a detailed clarification.
>
> For **(B) multi-fragment assembly**, our experiments show that our method can handle multiple parts (Appendix E.1). The relatively low success rate is mainly due to the presence of more minimal parts that are nearly impossible to grasp or assemble. Rather than excluding these highly challenging cases from test dataset, we intentionally include them for an honest evaluation. We recognize that multi-fragment assembly introduces additional challenges, we will explore pre-grasp and pre-orientation operations to handle such complexities and improve multi-fragment assembly.
>
> > W3. Quantitative results for real-world experiments. Sim2real transfer analysis.
>
> For (A) real-world experiments, we tested our model on each category with 10 trials, without any fine-tuning. The success rates are as follows: Bowl: 3/10, Mug: 2/10, BeerBottle: 3/10, WineGlass: 2/10. For mug, in some trials, we intentionally place it with the handle facing downward, making it ungraspable, so the gripper must grasp mug's top edge. This leads to collisions when both grippers grasp the top edges, due to mug's small diameter. For wineglass, its glasswork is prone to slipping, even when gripper successfully grasps it, it may tip during manipulation.
>
> For (B) sim2real transfer analysis, we load the object meshes (acquired from our real-world benchmark) into simulation. We observe better results in simulation, primarily due to discrepancies in joint constraints. For instance, when picking up a flat fragment on a table, gripper in simulation can move parallel and close to the table surface, whereas real robot encounters joint limitations that restrict its movement. This comparison highlights the need to incorporate bimanual joint constraints into simulation framework to better reflect real-world scenarios and improve transferability.
>
> > W4. floating grippers
>
> In this work, following [Paper 9-11], we focus on learning collaborative affordance, abstracting away the robot arm control. While our real-world experiments show the proposed actions work with real arms using motion planning (MoveIt!), we acknowledge that incorporating arm control would enhance the system’s realism. In future work, we will address these challenges including arm singularities and collision issues, integrating cuRobo for motion generation for bimanual manipulators.
>
> > W5. failure analysis and actionable insights
>
> In our failure analysis, we provided potential solutions for future works, including (1) incorporating pre-grasp operations like moving flat part to the table edge to make it graspable, (2) performing a series of pick-and-place operations to adjust object' pose. More details are available in Appendix G.
>
> > W6. Which component is the most brittle part?
>
> We have conducted additional ablation studies, with results and analysis in Table 4,5 of Appendix E.3.
>
> > W7. Is model robust to occlusions or noise?
>
> As described in Equation 2 of our paper, the pose estimator does not need to precisely predict the absolute object pose. Instead, it only needs to estimate the relative pose between two frames, which significantly simplifies the task. Beisdes, our empirical results show that FoundationPose, the SOTA model we use, performs well in continuous manipulation scenarios, maintaining accuracy and robustness even with occlusions (e.g., gripper occlusion after grasping) or sensor noise.
>
> **References**
>
> [9] Eisner, et al. Flowbot3d: Learning 3d articulation flow to manipulate articulated objects. RSS, 2022.
>
> [10] Xu, et al. UMPNet: Universal manipulation policy network for articulated objects. RAL, 2022.
>
> [11] Zhao, et al. Dualafford: Learning collaborative visual affordance for dual-gripper manipulation. ICLR, 2023.

---

### Official Review · Reviewer_KY79 · 2025-03-12

**Overall Recommendation:** 3

**Summary:**

In this paper, the authors present a framework for bimanual geometri assembly. They formulate the task into 3 steps: pick-up, alignment and assembly. For pick-up, a point-level affordance prediction module is trained and used; For alignment, a SE(3) transformation is predicted; For assembly, a collision-free direction is predicted.
The authors also introduce a real-world benchmark featuring geometry variety and global reproducibility.
The authors evaluate their method in simulation and on real data. The results outperform previous affordance-based and imitation-based methods.

**Claims And Evidence:**

The claims are well supported by the comparison and ablation studies results.

**Essential References Not Discussed:**

NA

**Experimental Designs Or Analyses:**

The authors compare their approach using three methods: ACT, a manually designed heuristic strategy, and a modified version of the existing affordance prediction method, DualAfford. However, they do not include comparisons with existing works on geometric assembly. The authors justify this omission by stating that these prior works do not account for robot execution. While this reasoning has merit, it would strengthen the study if the authors had adapted these geometric assembly methods to the current setting and included them in the comparison. Such an approach could provide a more comprehensive evaluation of their method’s performance relative to established baselines.
Additionally, the authors perform ablation studies to assess their design choices, which are reasonably constructed.

**Methods And Evaluation Criteria:**

The authors use the assembly success rate as the evaluation criteria, which can be reasonable. But the thresholds of distance and rotation angels are not given. It is very important for real applications. And according to the figure and video, there are still a large gap between the two parts after assembly.

**Other Comments Or Suggestions:**

NA

**Other Strengths And Weaknesses:**

The proposed framework is designed specifically to the task. However, its technical novelty is quite limited and does not meet the standards expected for ICML.
The paper is well-written, and the style of the figures is visually appealing. That said, the logic of Figure 2 is convoluted and difficult to follow.
Additionally, the method relies on complete reconstruction from multi-view images,which may be impractical for real-world applications.

**Questions For Authors:**

NA

**Relation To Broader Scientific Literature:**

The contribution of this work is very specific to this task, and may not contribute much to the broader community.

**Theoretical Claims:**

There are no theoretical claims.

---

> ### Author Rebuttal · Authors · 2025-04-01
>
> We sincerely appreciate your thoughtful questions. We've addressed them in detail below.
>
> **For detailed paper references, please refer to our response to Reviewer1 (z3UB).**
>
> > W1. Thresholds of distance and rotation. There is a gap between two parts after assembly.
>
> Thank you for the suggestion. The threshold for distance is set to 2 unit-length in the simulation, and threshold for rotation is 30 degrees. While we are not permitted to make revisions during rebuttal, we will include this information in the final paper. Geometric shape assembly is a particularly challenging task due to the diversity of object categories, complex geometries, and the significant generalization required. In fact, even humans find it difficult to assemble shapes based solely on visual information. In this paper, we have taken a successful first step toward addressing this challenge. Moving forward, we plan to incorporate tactile information into our method, as it is especially valuable in contact-rich tasks and may help achieve more precise assembly of fractured parts.
>
> > W2. Comparisons with existing works on geometric assembly.
>
> We conducted a comparison with an existing geometric assembly method [Paper-5]. Although this method only considers the geometries and ideal assembled poses of fractured parts, without taking the robotic assembly process into account, we adapted it as a baseline by: (1) using a heuristic method to generate the robots' pick-up actions (detailed in Appendix B), (2) denoting the predicted SE(3) pose for part i as $q_{i}^{asm}$, and using Equation 1 in our paper, we calculate the gripper’s target pose $g_{i}^{asm}$ for assembly. The average accuracy of this baseline is 3.00% on the training categories, which is significantly lower than our method. The main reason for this performance gap is that prior visual assembly methods neglect the robotic execution process. Specifically, these methods do not determine where to grasp the fragments, not only for successful pickup but also to avoid the seam region for subsequent assembly. They also lack the capability to align the fragments properly at the seam, which is crucial for avoiding collisions when the two parts are brought together. On the contrary, our method integrates the considerations of part geometry, shape assembly with robotic coordination and execution in the proposed affordance learning framework.
>
> > W3. The contribution of this work is specific to this task.
>
> Thank you for this constructive comment. Our framework focuses on learning bimanual collaborative and geometry-aware affordances to generate long-horizon action sequences for robotic manipulation tasks. While we demonstrate the effectiveness of our method in geometric shape assembly task, which involves diverse object categories, complex geometries, and significant generalization challenges, the learned affordance is applicable to a broader range of bimanual tasks that require coordinated manipulation.
>
> For other tasks requiring bimanual coordiniation, such as peg insertion, bottle cap closing, and furniture assembly (which are relatively easier than geometric shape assembly), our method can be easily adapted. For instance, in the bottle cap closing task, the process can be formulated into three steps: (1) The two arms pick up the bottle and cap from the table. (2) The two arms align the cap with the bottle opening. (3) The two arms place and secure the cap onto the bottle. We conducted experiments on this task, and results show that after training on a few bottle shapes, our method generalizes to novel bottle shapes, achieving an average accuracy of 67%. Visualizations of the predicted affordance maps and manipulation process are available on project website (Rebuttal-Figure3) [https://sites.google.com/view/biassembly/ ] .
>
> > W4. Figure 2 is convoluted
>
> In the paper, Figure 1 introduces the intuition behind our proposed framework, while Figure 2 provides a more detailed illustration. Based on your feedback, we have revised and simplified Figure 2 to make it clearer and easier to follow. The updated version is on our website (Rebuttal-Figure2).
>
> > W5. The method relies on reconstruction from multi-view images, which may be impractical for real-world applications.
>
> Reconstructing unknown objects (such as fractured parts) with robotic manipulators is a well-established research area. A feasible approach involves using two robotic arms with wrist camera to capture the images of fractured part and applying the method in Sec. 5.2 for reconstruction. The process and results are available at project website (Rebuttal-Figure1). Additionally, alternative approaches have been explored in previous works [Paper 1-3], including allowing the robot to re-orient the object while collecting visual observations to facilitate the reconstruction of unknown objects. Therefore, object reconstruction in real-world scenarios is both feasible and not inherently difficult, given the available techniques and tools.

---

> > ### Comment · Reviewer_KY79 · 2025-04-02
> >
> > Thank the author for the detailed and careful responses. They have conducted additional experiments, including new baseline and extension to other tasks. Most of my concerns have been addressed in the rebuttal. I'd like to raise my score to ``weak accept''. However, the technical novelty needs futher clarification in the final version.

---

> > > ### Author Response · Authors · 2025-04-02
> > >
> > > Dear Reviewer,
> > >
> > > We sincerely thank you for your thoughtful comments and for considering raising your score to "weak accept". We are pleased to hear that our responses have addressed most of your concerns. Following your suggestion, we will include further clarifications regarding the technical novelty (including our response to W3) and other discussions in the final version of the paper.
> > >
> > > We truly appreciate your positive recognition of our work. We would be very grateful if you could consider adjusting your score through the "edit" option on the original review if you find it appropriate.
> > >
> > > Once again, thank you for your valuable feedback and for your positive consideration.

---

### Official Review · Reviewer_64ad · 2025-03-14

**Overall Recommendation:** 3

**Summary:**

This work focuses the shape assembly task aimed at reconstructing broken objects. A multi-stage BiAssembly framework is put forward to carry out this task. Initially, the BiAssembly framework utilizes SOTA techniques to obtain an imagined assembled shape. Subsequently, it forecasts the disassembly direction, alignment pose transformation, pick-up affordance, and ultimately, the gripper alignment and assembly poses. Moreover, a real-world framework is introduced in this paper. The experimental outcomes demonstrate that the BiAssembly framework outperforms previous approaches.

**Claims And Evidence:**

Yes

**Essential References Not Discussed:**

Yes

**Experimental Designs Or Analyses:**

Yes

**Methods And Evaluation Criteria:**

Yes

**Other Comments Or Suggestions:**

Overall, this paper presents a viable framework for shape assembly, which is beneficial for this field. However, the framework in this paper is limited by its single-task-oriented design. This makes such methods less general and less likely to inspire a wider readers. I tend to accept this paper. Meanwhile, I hope that the author can strive for greater generality in the design of the model in the feature.

**Other Strengths And Weaknesses:**

Strengths
Overall, the paper is written well. Technical details and experiments are clearly explained. The framework for shape assembly seems to work. According to the paper's results, it does better than old heuristic or policy - based methods.

Weaknesses
The multi-stage framework has some assumptions. For example, it assumes the object has two broken parts, the imagined assembled shape can be got early, and the robot follows a set alignment and assembly process.

**Questions For Authors:**

None

**Relation To Broader Scientific Literature:**

This work is related to shape assemble perdition task. And it focus on the execution policy.

**Theoretical Claims:**

Yes

---

> ### Author Rebuttal · Authors · 2025-04-01
>
> Thank you for your valuable questions. We've addressed each of your concerns below.
>
> **For detailed paper references, please refer to our reply to Reviewer1 (Reviewer z3UB).**
>
> > W1. The multi-stage framework has some assumptions. For example, it assumes the object has two broken parts, the imagined assembled shape can be got early, and the robot follows a set alignment and assembly process.
>
> For **(A) multiple broken parts**. We have conducted experiments to demonstrate that our method can handle multiple broken parts in **Appendix E.1**. And both the quantitative and qualitative results demonstrate that our proposed method can be effectively adapted to multi-fragment assembly tasks. We also provide a detailed explanation of how our method can be adapted for multi-fragment assembly in this section.
>
> For **(B) imagined assembled shape**. The assumption of "imaginary assembled shape" is justified based on two well-established research areas that together ensure both adaptability and autonomy in real-world scenarios: (1) the reconstruction of broken parts, and (2) the prediction of target assembled shapes from those broken parts:
>
> 1. Reconstructing unknown objects (broken parts) with robotic manipulators is a well-studied problem [Paper 1-3]. A feasible approach involves using two robotic arms to capture images of the fractured part and applying the method in Sec. 5.2 for reconstruction. The process and results are available at project website (Rebuttal-Figure1) [ https://sites.google.com/view/biassembly/ ] . Additionally, alternative approaches have been explored in [Paper 1-3].
>
> 2. Predicting the imaginary assembled shape from multiple fractured parts is also a well-studied vision problem [Paper 4-8]. Prior works have demonstrated the ability to predict precise fragment poses and shown strong generalization capability to unknown parts and shapes, enabling the construction of an imaginary assembled shape. Furthermore, our experiments (Appendix E.2) show that our method is robust to imperfect imaginary assembled shapes, even without fine-tuning.
>
> These supporting works and empirical results demonstrate the adaptability and autonomy of our framework in real-world scenarios.
>
> For **(C) alignment and assembly process**. The alignment and assembly process mirrors the natural approach humans take when assembling fragments. Humans typically align the fragments along the seams first and then gradually move them together for precise fitting. Furthermore, when decomposing the assembly process into multiple frames, there is usually a stage where the two fragments are aligned but separated by a small distance. This intermediate step is captured in our formulation as the alignment step, which generalizes well to most shape assembly scenarios.
>
> Thanks for your valuable comments! We will add the above discussions in our paper and make it more clarified.
>
>
>
> > W2. Overall, this paper presents a viable framework for shape assembly, which is beneficial for this field. However, the framework in this paper is limited by its single-task-oriented design. This makes such methods less general and less likely to inspire a wider readers. I tend to accept this paper. Meanwhile, I hope that the author can strive for greater generality in the design of the model in the feature.
>
> Thank you for this constructive comment. Our framework focuses on learning bimanual collaborative and geometry-aware affordances to generate long-horizon action sequences for robotic manipulation tasks. While we demonstrate the effectiveness of the proposed method in geometric shape assembly task,  which involves diverse object categories, complex geometries, and significant generalization challenges, the learned affordance is applicable to a broader range of bimanual tasks that require coordinated manipulation.
>
> For other tasks requiring bimanual coordiniation, such as peg insertion, bottle cap closing, and furniture assembly (which are relatively easier than geometric shape assembly), our method can be easily adapted. For instance, in the bottle cap closing task, the process can be formulated into three steps: (1) The two arms pick up the bottle and cap from the table. (2) The two arms align the cap with the bottle opening. (3) The two arms place and secure the cap onto the bottle. We conducted experiments on this task, and the results show that after training on a few bottle shapes, our method generalizes to novel bottle shapes, achieving an average accuracy of 67%. Visualizations of the predicted affordance maps and the manipulation process are available on our project website (Rebuttal-Figure3)  [ https://sites.google.com/view/biassembly/ ] .

---

> > ### Comment · Reviewer_64ad · 2025-04-02
> >
> > Thank you for the author's response. I will maintain my initial rating to support the acceptance of this paper.

---

> > > ### Author Response · Authors · 2025-04-03
> > >
> > > Dear reviewer,
> > >
> > > We are pleased that our clarifications have addressed your concerns. Thanks for your positive rating and recommendation to acceptance!

---

### Official Review · Reviewer_z3UB · 2025-03-16

**Overall Recommendation:** 3

**Summary:**

This paper addresses the challenges in the observation space and action space by proposing the BiAssemble framework to solve the collaborative problem of bimanual robots in geometric assembly tasks. Specifically, the task is decomposed into three steps: pick-up, alignment, and assembly, which are addressed by progressively predicting the affordance maps and the gripper actions of the two parts of the object. Additionally, this paper establishes a real-world benchmark for assembling broken objects and conducts extensive experiments in both simulation and real-world environments, demonstrating the superiority of the proposed algorithm and its ability to generalize to unseen object categories.

**Claims And Evidence:**

clear

**Essential References Not Discussed:**

no

**Experimental Designs Or Analyses:**

yes

**Methods And Evaluation Criteria:**

yes

**Other Comments Or Suggestions:**

none

**Other Strengths And Weaknesses:**

Strengths:
•	The paper proposes a novel geometric assembly task focusing on bimanual robot collaboration to repair broken objects, addressing the research gap in the field of geometric assembly for complex shapes and long-horizon action sequences.
•	The idea of introducing point-level affordance is both interesting and highly significant, as it not only predicts the feasibility of grasping points but also simultaneously considers the collaborative requirements of subsequent alignment and assembly steps.
•	The construction of a real-world benchmark bridges the gap between simulation and real-world environments.
•	The article is well-written, with a clear structure, well-defined motivations, and comprehensive experiments.
Weaknesses:
•	The algorithm relies on predefined assembly shapes, and I suspect that it is unable to autonomously infer the correct assembly of unknown objects, thus limiting its applicability and reducing its adaptability and autonomy in real-world scenarios.
•	The paper assumes that the relative pose between the gripper and the object remains stable, but in real environments, the relative attitude between the gripper and the object may change due to external perturbations. Does this perturbation significantly affect the robustness of the model? If so, in what ways? If the perturbation significantly affects the model performance, in what ways are subsequent plans to address this issue?
•	For the implementation of the methodology mentioned in the authors' supplementary material that can be extended to handle multi-piece assemblies, how does the accumulation of iterative errors affect the results, and will an end-to-end approach be considered for efficient completion in multi-piece tasks?

**Questions For Authors:**

see Strengths And Weaknesses

**Relation To Broader Scientific Literature:**

none

**Theoretical Claims:**

yes

---

> ### Author Rebuttal · Authors · 2025-04-01
>
> We sincerely appreciate your valuable questions, and we have provided detailed responses below.
>
> > W1. The algorithm relies on predefined assembly shapes...
>
> The assumption of "imaginary assembled shape" is justified based on two well-established research areas that together ensure both adaptability and autonomy in real-world scenarios: (1) the reconstruction of broken parts, and (2) the prediction of target assembled shapes from those broken parts:
>
> 1. Reconstructing unknown objects (broken parts) with robotic manipulators is a well-studied problem [Paper 1-3]. A feasible approach involves using two robotic arms to capture images of the fractured part and applying the method in Sec. 5.2 for reconstruction. The process and results are available at project website (Rebuttal-Figure1) [https://sites.google.com/view/biassembly/ ]. Additionally, alternative approaches have been explored in [Paper 1-3].
> 2. Predicting the imaginary assembled shape from multiple fractured parts is also a well-studied vision problem [Paper 4-8]. Prior works have demonstrated the ability to predict precise fragment poses and shown strong generalization capability to unknown parts and shapes, enabling the construction of an imaginary assembled shape. Furthermore, our experiments (Appendix E.2) show that our method is robust to imperfect imaginary assembled shapes, even without fine-tuning.
>
> These supporting works and empirical results demonstrate the adaptability and autonomy of our framework in real-world scenarios.
>
> > W2. The relative attitude between gripper and object may change due to external perturbations. Does it affect the robustness of model?
>
> If external perturbations cause the relative pose between the gripper and the object to change, a pretrained pose estimation model (we used FoundationPose) can track the updated object pose in real-time performance. Consequently, as shown in Equation 2 of our paper, we only need to update the previous gripper pose $ q_{i}^{pick} $ with the perturbed pose $ \hat{q_{i}^{pick}} $, allowing us to compute the correct gripper pose $ \hat{g_{i}^{asm}} $ for assembly. Furthermore, since Equation 2 holds at any time step, our approach remains robust throughout the manipulation process. Specifically, at each time step $ t $, we update $ q_{i,t}^{pick} $ (obtained from FoundationPose) and $ g_{i,t}^{pick} $ (obtained from the robot control interface) to compute the appropriate gripper pose $ g_{i,t+1}^{pick} $ for the next step. This ensures that our method can dynamically adapt to perturbations without compromising assembly accuracy. We appreciate this insightful question and will incorporate this explanation into our paper after rebuttal.
>
> > W3. How does the accumulation of iterative errors affect the results of multi-piece task? Will an end-to-end approach be considered for efficient completion in multi-piece tasks?
>
> To evaluate the impact of iterative error accumulation in multi-piece assemblies, we conduct a comparative experiment with (a)(b) settings in the first iteration: (a) the two parts are assembled by the robot, (b) the two parts are perfectly assembled using ground-truth alignment. Then, we evaluate the accuracy of the second-iteration assembly ''to assemble the third parts to'' under (a)(b) conditions: 21.20% accuracy for setting (a), and 24.80% accuracy for setting (b). These results indicate that iterative errors affect assembly accuracy, as misalignments in earlier steps can propagate and influence the integration of new parts.
>
> We appreciate the suggestion regarding an end-to-end approach. While our current method is effective for multi-piece assembly, we will explore an end-to-end approach in future work. This would not only consider the geometry of the parts being assembled in each iteration but also optimize the overall assembly sequence based on the geometry of all parts, leading to more efficient and accurate multi-piece assembly.
>
>
>
> **References**
>
> [1] Nicholas Pfaff, et al. Scalable Real2Sim: Physics-Aware Asset Generation Via Robotic Pick-and-Place Setups. 2025
>
> [2] Saptarshi Dasgupta, et al. Uncertainty-aware Active Learning of NeRF-based Object Models for Robot Manipulators using Visual and Re-orientation Actions. IROS, 2024.
>
> [3] Zhizhou Jia, et al. An Efficient Projection-Based Next-best-view Planning Framework for Reconstruction of Unknown Objects. 2025.
>
> [4] Silvia Sellán1, et al. Breaking bad: A dataset for geometric fracture and reassembly. Neurips, 2022.
>
> [5] Ruihai Wu, et al. Leveraging SE-(3) equivariance for learning 3d geometric shape assembly. ICCV, 2023.
>
> [6] Jiaxin Lu, et al. Jigsaw: Learning to Assemble Multiple Fractured Objects. Neurips, 2024.
>
> [7] Theodore Tsesmelis, et al. Re-assembling the past: The RePAIR dataset and benchmark for real world 2D and 3D puzzle solving. Neurips, 2024.
>
> [8] Gianluca Scarpellini, et al. DiffAssemble: A Unified Graph-Diffusion Model for 2D and 3D Reassembly. CVPR, 2024.

---

### Decision · Program_Chairs · 2025-05-01

**Decision:**

Accept (poster)

**Comment:**

This paper received four weak accepts after rebuttal. All reviewers acknowledged its novel contribution to geometric assembly tasks, addressing key challenges in complex shape manipulation and long-horizon action planning. The proposed point-level affordance method was noted as both innovative and practically relevant. The paper is well-structured, with clear motivations and comprehensive experimental validation. Most concerns raised during review were resolved in the rebuttal; notably, one reviewer raised their score from 2 to 3, solidifying the consensus. The Area Chair concurs with the reviewers and recommends Accept.